



**Shallow boundary layer heights controlled by the surface-based temperature inversion**
**strength are responsible for trapping home heating emissions near the ground level in**
**Fairbanks, Alaska.**
**Authors:** Meeta Cesler-Maloney[1], William R. Simpson*[1], Jonas Kuhn[2], Jochen Stutz[2], Jennie L.
Thomas[3], Tjarda Roberts[4], Deanna Huff[5] and Sol Cooperdock[2]
**Affiliations:** [1] Geophysical Institute and Department of Chemistry and Biochemistry, University
of Alaska Fairbanks, Fairbanks, AK 99775
[2] UCLA Atmospheric & Oceanic Sciences, Los Angeles, CA 90095
[3] Univ. Grenoble Alpes, CNRS, INRAE, IRD, Grenoble INP, IGE, 38000 Grenoble, France
[4] LMD/IPSL, ENS, Université PSL, École Polytechnique, Institut Polytechnique de Paris,
Sorbonne Université, CNRS, Paris, France
[5] Alaska Department of Environmental Conservation, P.O. Box 111800, Juneau, 99811-1800
*Corresponding author: William Simpson, wrsimpson@alaska.edu
**Abstract**
In cold climate cities, like Fairbanks, Alaska, during winter, reduced vertical mixing in the
atmosphere leads to pollution trapping and concerningly high $PM_{2.5}$ concentrations at ground level.
To study pollution trapping, we simulated dispersion of $SO_2$ from home heating emissions during
the ALPACA-2022 field study in Fairbanks, Alaska using the Platform for Atmospheric Chemistry
and Transport one-dimensional model (PACT-1D). Eddy diffusion coefficients that control
vertical transport were parameterized by the near-surface temperature inversion strength according





to stable boundary layer (SBL) theory and horizontal export was calculated from the wind speed.
The model parameterized the SBL height as a function of the near-surface inversion strength, with
the SBL height varying between 50 m for weak inversions down to 20 m for strong inversions.
The model results were compared to long-path differential optical absorption spectroscopy (LP-
DOAS) concentration profiles and in-situ observations of $SO_2$ over the range of 3 m to 191 m
above downtown Fairbanks over a 33-day period in winter and achieved excellent agreement ($R =$
0.88). Sensitivity studies showed that the model is most sensitive to the SBL height and the
associated eddy diffusivity profile. Model-derived pollution residence times in Fairbanks are on
the order of hours during winter, with a median steady state residence time of 2.1 hours under
stable atmospheric conditions, indicating there is limited time for chemical processing.
**Plain language summary (Short summary)**
We used a one-dimensional model to simulate how pollution in Fairbanks, Alaska,
accumulates in shallow layers near the ground when temperature inversions are present. We find
pollution accumulates in a 20 m to 50 m thick layer. The model agrees with observations of $SO_2$
pollution using only home heating emissions sources, which shows that ground-based sources
dominate sulfur pollution in downtown Fairbanks. Air residence times in downtown are only a few
hours, limiting chemical transformations.
**Key points**
1. The PACT-1D model was used to model pollution in Fairbanks, Alaska, using vertical and
horizontal dispersion.
2. Observed path averages of sulfur dioxide measured by LP-DOAS from 12 m to 191 m in
downtown Fairbanks for 33 days in winter 2022 were modeled with good skill ($R = 0.88$).





3. The median pollution residence time was 2.1 hours when the atmosphere was stable, limiting time for chemical processing.

**Keywords:** one-dimensional modeling, long-path differential optical absorption spectroscopy, vertical profiling, dispersion, Alaska, Fairbanks, pollution trapping, temperature inversion, cold climate

**1 Introduction**

Weak atmospheric dispersion combined with local emissions sources causes many cold-climate cities to have poor air quality during winter (ALPACA, 2018; Schmale et al., 2018). This poor dispersion is often caused by surface-based temperature inversions, which hinder vertical mixing and prevent winds aloft from penetrating to ground level, where polluted air stagnates. Higher-latitude locations like Fairbanks, Alaska experience frequent surface-based temperature inversions during winter. The Fairbanks North Star Borough is designated by the Environmental Protection Agency (EPA) as a non-attainment area for fine particulate matter, $PM_{2.5}$, which episodically exceeds the 24-hour standard of 35 microgram $m^{-3}$ during winter. While Fairbanks does not exceed the national ambient air quality standards for $SO_2$, ground level $SO_2$ mixing ratios can reach 1-hour averages up to 40 nanomole mole$^{-1}$, sometimes for multiple hours (ADEC, 2016). Improvement of air quality in Fairbanks has been a high priority for stakeholders, but our limited understanding of the sources, chemical transformations, and dispersion of pollutants under stagnant winter conditions has hindered these efforts.

Fairbanks residents rely on a variety of fuel sources less commonly used in warmer regions, including wood (both cordwood and pellets) and No. 1 or No. 2 heating oil, both of which have



orders of magnitude higher sulfur content compared to the natural gas and/or ultra-low sulfur diesel
(heating oil) that is more widely used in the contiguous USA. Chemical mass balance modeling
over a three-winter period from 2008 through 2011 attributed 60% to 80% of $PM_{2.5}$ mass
concentration to wood burning in downtown Fairbanks (Ward et al., 2012). A positive matrix
factorization analysis by Kotchenruther et al. (2016) attributed 51.8% of $PM_{2.5}$ mass concentration
to wood burning in downtown Fairbanks. Recent PMF analysis indicates that the fractional
contribution of wood to $PM_{2.5}$ in downtown Fairbanks is trending to smaller amounts in recent
years (Ye and Wang, 2020).

Sulfate is the second most prevalent species in $PM_{2.5}$, so understanding sulfur sources and

chemistry is important to Fairbanks air quality (Ward et al., 2012). The observed wintertime sulfur
oxidation ratio, or the ratio of observed moles of sulfate to total moles of atmospheric sulfur
(combined $SO_2$ and sulfate), is about 5%, which suggests that although secondary chemistry may
be adding some sulfate to $PM_{2.5}$, 95% of the sulfur emitted remains in the gas phase as $SO_2$
(Nattinger, 2016). This low sulfur oxidation ratio is in agreement with recent analysis of sulfate
isotopes in Fairbanks during winter, which showed that 62% ±12% of sulfate came from primary
sources, with a smaller influence of secondary chemistry (Moon et al., 2024). In addition to home
heating sources, vehicles emit CO, $PM_{2.5}$ and $NO_x$ at ground level. Six power plants fueled by
coal, diesel, and naphtha, referred to here as point sources, emit large amounts of pollutants. The
power plants have tall stacks intended to push their exhaust above the surface-based temperature
inversion (ADEC, 2016). While not an air quality regulated pollutant, carbon dioxide ($CO_2$) is co-
emitted by all $SO_2$ emission sources in Fairbanks and also sourced from mobile sources and other
heat sources. Because it does not undergo chemistry and is not consumed by plants in winter, it





can serve as an important marker for combustion processes, as well as the dispersion of these
emissions.

While local emissions cause the poor wintertime air quality in Fairbanks, dispersion

processes also significantly influence the amount of pollution that accumulates at breathing level.
Tran and Mölders (2011) found that $PM_{2.5}$ concentrations in Fairbanks were larger when there
were surface-based temperature inversions and stagnant winds near the ground, concurrent with
low temperatures, which increase ground level home heating emissions. Surface-based inversions
are often intense, with temperature gradients of ~1°C m$^{-1}$ or more observed in the first few meters
above the ground level (AGL) on the flat valley floor and can persist throughout the day and even
for multiple days during winter (Benson, 1969; Bowling, 1986; Tran and Mölders, 2011; Mayfield
et al., 2013; Cesler-Maloney et al., 2022). With the weak vertical dispersion and stagnant
horizontal wind speeds near the ground during surface-based inversions, emissions from ground
level sources are thought to have a larger impact on ground level pollution in Fairbanks than power
plant emissions, although this contribution has only been investigated using numerical models and
has not yet been validated by field observations (ADEC, 2016; Tran and Mölders, 2012). This
previous work in Fairbanks showed that pollutants were present at higher concentrations during
stagnant conditions, indicating that dispersion is crucial to controlling the amount of pollution that
accumulates near ground level.

There are two distinct mechanisms responsible for the development of surface-based

temperature inversions in Arctic regions: radiative cooling at the surface and warm air advection
(Busch et al., 1982; Bowling, 1986; Bradley et al., 1992; Bourne et al., 2010; Zhang et al., 2011).
During radiative cooling, heat is lost by infrared radiation from the ground, and air near the ground
that interacts with this cooling surface decreases in temperature more than the air at higher



altitudes. These radiative cooling inversions become more intense on nights with clear sky
conditions. The formation of surface-based inversions during winter in Fairbanks is also supported
by the high emissivity of snow in the thermal infrared and the high albedo of snow in the visible,
limiting solar heating. When the warm air advection mechanism occurs, warmer, more buoyant,
air masses are advected over colder, denser, air masses laying near the surface, creating a surface-
based temperature inversion.

During surface-based inversions, the atmosphere is stable with regard to buoyancy, which

reduces turbulence, but mechanical friction of wind acting on the surface roughness leads to some
mixing. This stable part of the atmosphere located near the ground level during surface-based
inversions is referred to as the stable boundary layer (SBL). On short time and length scales, this
mixing is highly complex, including intermittency and often showing oscillations (Mahrt, 1981);
however, simpler 1-D empirical theories are able to describe effective turbulent transport in the
SBL with good agreement to observations on longer timescales or via spatial averaging. Monin-
Obukhov similarity theory defines a set of functions to describe the vertical mean flow and
temperature in the surface layer as a function of dimensionless height, which is related to actual
height using the Obukhov length parameter, $L_{MO}$ (Monin and Obukhov, 1954; Mahrt, 1989). The
Obukhov length is a key parameter in many of these turbulence models, along with the SBL height,
$h$, which is the height above which the atmosphere is no longer influenced by contact with the
surface. Vertically mixing turbulent eddies exist between the ground surface and the top of this
SBL, but there is little vertical exchange in the free troposphere above the SBL height. Past work
has shown that the SBL height decreases with increased surface cooling rate, sometimes to SBL
height of just tens of meters (Mahrt, 1981; Wyngaard, 1975; Brost and Wyngaard, 1978; Stull,
1983; Nieuwstadt, 1984a, b).



Empirical theories that describe bulk turbulence in the SBL parameterize vertical mixing
through the eddy diffusivity coefficient for momentum ($K_z$) (Newsom and Banta, 2003;
Nieuwstadt, 1984b; Wyngaard, 1975; Brost and Wyngaard, 1978; Beare et al., 2006; Degrazia and
Moraes, 1992). These models have a $K_z$ vertical profile that peaks at an altitude that is roughly
20% of the SBL height and decreases to the molecular diffusion limit (near zero) both at the surface
and above the top of the SBL (Kuhn et al., 1977; Nieuwstadt, 1984a; Brost and Wyngaard, 1978;
Degrazia and Moraes, 1992; Beare et al., 2006). In this work, we will use these models for the $K_z$
vertical profile to describe vertical mixing within the SBL.
Here, the Platform for Atmospheric Chemistry and Transport one-dimensional (PACT-1D)
atmospheric column model (Tuite et al., 2021; Ahmed et al., 2022) was used to analyze how the
dispersion of pollution depends on meteorological conditions during winter in Fairbanks. The
model simulates vertical turbulent mixing with high vertical resolution and uses horizontal
exchange with the unpolluted ambient atmosphere driven by winds to simulate dispersion of $SO_2$
emissions in downtown Fairbanks. Model results were validated against path-averaged field
observations of $SO_2$ measured by a long-path differential optical absorption spectroscopy (LP-
DOAS) instrument deployed in downtown Fairbanks as part of the Alaskan Layered Pollution And
Chemical Analysis (ALPACA) study, which was carried out in Fairbanks during January and
February 2022 (Simpson et al., submitted 2023). Model results were also compared with in-situ
$SO_2$ observations at 3 m AGL and $CO_2$ measurements at 3 m and 23 m AGL. PACT-1D simulations
were also used to calculate the transport loss rates of pollution from both vertical and horizontal
dispersion to understand the residence time of pollution in the urbanized area of Fairbanks. Having
an estimate of the residence time of pollution in the city helps to provide a constraint on the
timescales for chemical processing in Fairbanks. With a better understanding of these dispersion





processes, communities and regulatory agencies can better predict the timing and severity of
pollution events and develop improved strategies for pollution mitigation based on meteorological
forecasts.

**2 Field measurements from the ALPACA campaign**
The ALPACA field campaign took place at multiple sites in Fairbanks in winter, from
January 17 to February 25, 2022. Fairbanks, located in central Alaska near the Arctic circle, is one
of the few urbanized cities in the region and lies within a basin, surrounded by hills to the north,
east and west, with flat lands to the south. Fairbanks winters are characterized by short daylight
hours and extremely cold temperatures with frequent surface-based inversions.
Measurements used in this manuscript were made in downtown Fairbanks at two sites: the
University of Alaska Fairbanks Community and Technical College field site (CTC site, 64.841°N,
−147.727°E) and at the LP-DOAS base at 17 m AGL on the top floor of the Lacey-Street Parking
Garage (64.844°N, −147.716°E), which lies 610 m east-northeast of the CTC site. Figure 1 shows
a map of the relevant ALPACA measurement locations in Fairbanks. The nine boxes shown in
Figure 1 are the Weather Research Forecast / Community Multiscale Air Quality model
(WRF/CMAQ) 1.33 km-scale grid cells that were co-added to be the emissions footprint of PACT-
1D, as described in Section 3. The emissions inventory and WRF/CMAQ model was developed
by the U.S. Environmental Protection Agency (EPA) and the Alaska Department of Environmental
Conservation (ADEC). The LP-DOAS measured path-averaged trace gas mixing ratios across four
slightly inclined paths viewing north-east towards Birch Hill, a ~200 m tall hill. The shortest path
was ~1 km to the north-east from the base to a retroreflector situated at 12 m AGL on the Nordale
Elementary School building. The other three paths were roughly 4 km long and had retroreflectors





at altitudes of 73 m, 115 m and 191 m AGL at the LP-DOAS base. Nearly all of the populated
portion along these paths was within a few meters above the flat valley floor. The LP-DOAS
instrumentation used in the ALPACA field campaign is similar to the instrumentation used in past
studies to measure vertical profiles of UV-absorbing trace gases in urban environments and uses
analytical methods described in Platt and Stutz (2008) (Stutz et al., 2004; Tsai et al., 2014).

Temperature was measured at 3 m, 6 m, 11 m and 23 m AGL using aspirated thermometers

(Cesler-Maloney et al., 2022). The 3 m, 6 m and 11 m temperature probes were deployed on the
tower of a small stationary trailer in the parking lot of the CTC site, while the 23 m temperature
probe was deployed above the elevator shaft house roof on top of the CTC building. An RM Young
1005 wind speed and direction monitor was co-located with the 23 m temperature sensor. The
relative precision of these temperature sensors was better than 0.15 °C over the range from 20 °C
to −60 °C (Cesler-Maloney et al., 2022).

In-situ $SO_2$ (Thermo Scientific 43C) was measured from an inlet at 3 m AGL in a larger

stationary trailer parked next to the CTC building. The $SO_2$ analyzer was calibrated using an EPA
certified mixed standard containing 5.190 micromole mole$^{-1}$ $SO_2$ and 508.4 micromole mole$^{-1}$ CO
by overflowing the inlet with zero air or standard gas diluted in zero air at multiple calibration
mixing ratios using an Environics 9100 calibration dilution system. The gas analyzer was
calibrated roughly weekly and this multi-point calibration slope and zero were applied to the data
by linearly interpolating them in time between calibrations. Two Vaisala (GMP343) $CO_2$
instruments were also deployed at the CTC site, one at 3 m AGL on the small trailer and the other
at 23 m AGL on the roof of the CTC building. The $CO_2$ instruments were co-located at the same
altitude on the small trailer both before and after the field campaign and the older instrument was
corrected to the newer instrument that had a recent factory calibration within the prior year. Trace



gases ($O_3$, carbon monoxide (CO) and $NO_x$) were also measured at the CTC site and $O_3$ was
measured by a second instrument at 158 m AGL on Birch Hill (see the Supplemental Materials for
more information).

**3 Methods**
**3.1 Description of the PACT-1D model**

One-dimensional (1-D) chemical transport models simulate coupled dispersion and

chemical processes to predict the vertical profiles of pollutants. These 1-D models have simulated
vertical profiles of trace gases in urban areas, such as Houston, Texas and Los Angeles, California,
with good agreement to observations (Geyer and Stutz, 2004; Stutz et al., 2004; Tsai et al., 2014;
Tuite et al., 2021). Vertical and horizontal transport processes can both be simulated within the 1-
D model's column. Horizontal transport can be considered as an entrainment of background air
from outside of a polluted region into the modeled column layers. Fairbanks, being a small city
encircled by the sparsely populated boreal forest, is well suited for 1-D modeling with added
horizontal dilution by surrounding background air. This horizontal transport process is
implemented in the model by adding a pollution exchange term proportional to the windspeed
divided by the length scale of the urban area (see Section 3.3).

Here, we use a vertical grid, which divides the lower atmosphere (up to 1000 m AGL) into

39 vertical layers that vary in thickness, being denser near the ground. The PACT-1D model
simulates vertical exchange through eddy flux-gradient formalism, using vertical eddy diffusion
coefficients, $K_z$, for each layer.

The PACT-1D model was used in a dispersion-only mode, without simulating chemical

processes and $SO_2$ is used as a dispersion tracer of Fairbanks pollution. We chose $SO_2$ because it



has a well quantified and spatially distributed ground level source from home heating oil
combustion and its vertical distribution was measured by LP-DOAS during ALPACA, which is
used to validate the model results. Although $SO_2$ can be lost via chemical oxidation, the observed
sulfur oxidation ratio during Fairbanks winter is small, with only 5% of the total atmospheric sulfur
appearing as sulfate in observations (Nattinger, 2016). In addition to $SO_2$, $CO_2$ was also modeled
as a passive tracer of dispersion and validated against in-situ observations of $CO_2$ from the CTC
building site in Figure 1.

**3.2 Treatment of vertical exchange**
It is difficult to model vertical exchange in the shallow SBL, so we used a simple model
from literature to determine the vertical profile for $K_z$. Input $K_z$ profiles for PACT-1D were
calculated under SBL conditions using Equation EQ1 from Brost and Wyngaard (1978), where $h$
is the SBL height, $z$ is the altitude and $L_{MO}$ is the Monin-Obukhov length.
$$K_z(z) = \kappa u^* h \frac{(z/h)(1-z/h)^{1.5}}{1+4.7(z/h)(h/L_{MO})}$$
                                                                                                                                                    **EQ1**

In Equation EQ1, constant values were used for the von Kármán constant, $\kappa = 0.4$, the friction
velocity, $u^* = 0.40$ m s$^{-1}$, and the ratio of $h/L_{MO} = 1.4$. Brost and Wyngaard showed that variations
in $h/L_{MO}$ had only small effects on the vertical shape and peak of the $K_z$ profiles calculated by
Equation EQ1.
With these assumptions, the $K_z$ profile only depends upon the SBL height, $h$. For simulation
of the ALPACA field campaign period, the SBL height was parameterized as a function of the
near-surface atmospheric stability, which we quantify using the 23 m minus 3 m temperature
difference at the CTC site, d$T_{[23m-3m]}$. When d$T_{[23m-3m]}$ is large and positive, the atmosphere is very
stable, and when it is near zero or slightly negative, the atmosphere is neutral. Past work



demonstrates that when stability is very strong, the SBL height is shallow (Mahrt, 1981;
Wyngaard, 1975; Brost and Wyngaard, 1978; Stull, 1983; Nieuwstadt, 1984a, b). Conversely, as
d$T_{[23m-3m]}$ approaches neutral conditions (near zero), the SBL height is expected to increase. Based
upon this relationship between d$T_{[23m-3m]}$ and stability, a lookup table was created to calculate $h$
from d$T_{[23m-3m]}$, which was linearly interpolated between nodes in Table 1. Section 5.2 will discuss
the relationship between the SBL height, $h$, and the surface-based temperature inversion strength,
d$T_{[23m-3m]}$ in Table 1 in more detail.

Sensitivity studies were used to determine how model results depend upon these

parameterized SBL heights. When the atmosphere was near neutral (d$T_{[23m-3m]}$ < 0.2 °C), Equation
EQ1 was used with a value $h_{neutral}$ = 400 m, representing a tall SBL height. Figure S2 in the
Supplemental Materials shows that the resulting $K_z$ profile in the critical near-surface region is
nearly the same whether $h_{neutral}$ = 400 m, a larger SBL height or even a neutral $K_z$ profile is chosen.
Sensitivity studies will test the model result's dependence on this choice of a near-neutral SBL
height.


**3.3 Treatment of horizontal exchange**

The PACT-1D model was used in the configuration described in Tuite et al. (2021), but

with an added horizontal dispersion term, as described in Equation EQ2.

$$K_{EXCH} = v \, / \, L$$                          **EQ2**

Equation EQ2 represents the standard mass-balance approach result for the dependence of the
horizontal exchange coefficient, $K_{EXCH}$, on wind speed, $v$, and the length of Fairbanks, $L$ (Jacob,
1999). Here, a horizontal characteristic length scale of $L$ = 4 km is the length of the footprint of





the model column and was meant to represent the urban core of Fairbanks (Figure 1). The wind
speed at each model layer altitude is used to calculate a layer-specific exchange rate. Sensitivity
studies are used to test the model's response to varying this horizontal length parameter.

Constant wind speed profiles were used in idealized (steady state) model simulations, while

a combination of WRF-modeled and observed wind speeds were used as model inputs in
simulations of observed pollution events. Wind speeds from the WRF model were provided by the
U.S. EPA, which was sampled at the middle of the modeled area at 64.842°N, -147.700°E.
Observed wind speeds were measured at 23 m at the CTC site and also at 2 m and 10 m at the
nearby ADEC NCore site, located roughly 500 m north of the CTC site (Figure 1). When observed
winds were used as model inputs, the wind speed was linearly interpolated between 2 m (NCore),
10 m (NCore), 23 m (CTC) and 50 m (WRF) and only WRF winds were used above 50 m AGL.
The WRF wind speeds were compared to radiosonde data from the Fairbanks Airport and were
found to be in good agreement to radiosonde wind speeds in the first 500 m AGL.



**3.4 Treatment of emissions**

Emissions for Fairbanks were provided by the ADEC as hourly total emissions for the sum

of nine WRF/CMAQ (1.33 km resolution) grid cells arranged in a 3x3 grid cell = 4 km x 4 km =
16 km$^2$ area in downtown Fairbanks. This emissions area has a bounding box with NW corner:
64.861°N, -147.743°E and SE corner = 64.824°N, -147.658°E and includes the NCore site, the
CTC site, the A-Street site, the LP-DOAS base station and nearly all of the LP-DOAS paths up
Birch Hill, as shown in Figure 1. Downtown emissions rates were converted to moles m$^{-2}$ s$^{-1}$ using





the footprint of this emission area. Because PACT-1D has no horizontal dimensions, emissions
sources are instantaneously mixed throughout the layer in which they are emitted, such that the
model would not be able to capture the horizontal variability in emissions if sources were not
homogenously distributed across the area being modeled. However, $SO_2$ is emitted from hundreds
of homes within the 16 $km^2$ area in downtown Fairbanks, providing a distributed source that may
make the model's assumption of horizontal mixing within the column be more realistic.

The main source of $SO_2$ emissions near ground level is from home heating, as most homes

in Fairbanks relied on sulfur-rich No. 2 or No. 1 heating oils at the time of the ALPACA field
campaign, thus only home heating emissions data was used in model simulations. Pollutant
emissions from home heating sources were distributed into PACT-1D by altitude, with 40% in the
PACT-1D layer from 3 m to 6 m, 50% in the PACT-1D layer from 6 m to 9 m and 10% PACT-
1D layer from 9 m to 12 m. Carbon dioxide ($CO_2$) is produced in many combustion processes,
allowing $CO_2$ to be used as a tracer for anthropogenic pollution in the atmosphere. The ADEC
does not report $CO_2$ emissions, so an empirical emissions ratio of $CO_2$ to $SO_2$ was used to
determine the emissions rate for $CO_2$ in the model. A linear fit for observed 3 m $CO_2$ versus $SO_2$
at CTC was calculated, where the intercept is interpreted to be the $CO_2$ background mixing ratio
surrounding Fairbanks (420 micromole mole$^{-1}$) and the slope is the empirical emissions ratio of
4300 moles $CO_2$ to 1 mole $SO_2$, such that multiplying the $SO_2$ emissions ratio by this slope yields
the $CO_2$ emissions rate in the model (see Supplemental Materials Figure S1).

**3.5 Diagnosing vertical and horizontal exchange**

To diagnose the transport rate of $SO_2$ by vertical and horizontal dispersion in PACT-1D,

two loss rates were determined; the vertical loss rate out of the "urban canopy", defined as the



layer from zero to 15 m AGL (approximate height of trees and buildings) and denoted as $L_{canopy}$,
and the vertically integrated horizontal loss rate across the model column, denoted as $L_{export}$. The
PACT-1D model calculates vertical transport exchange rates for each layer (denoted $VT_{layer}$),
where positive rates represent net gain of concentration of a species in that layer by vertical
transport from neighboring layers and negative rates a net concentration loss. Equations EQ3 and
EQ4 show the transport loss rates calculated from model output.
$$L_{canopy} = \sum_{layers\ 0m-15m}(-VT_{layer} \times layer\ height) \qquad \textbf{EQ3}$$
$$L_{export} = \sum_{all\ layers}\left(K_{exch\_layer} \times layer\ height \times \left([C]_{layer} - [C]_{bg}\right)\right) \qquad \textbf{EQ4}$$

In this equation, $[C]$ represents the concentration of a species in each layer or in the

background air that replaces horizontally exported air. Both loss rates have units of column density
per time, mole m$^{-2}$ s$^{-1}$. Because the considered sources emit only within the urban canopy layer
and background air is cleaner than the polluted column being considered, both rates are positive,
indicating vertical transport upwards, out of the urban canopy, and export of pollution horizontally
out of the considered polluted urban column.


**3.6 Diagnosing steady state residence times for vertical and horizontal loss processes**

The steady-state lifetime of a box model is defined by Jacob (1999) as the ratio of the

amount of any species in the box divided by the total loss rate for that species out of the box. As
chemistry was not considered in this application, we refer to our box-model lifetime outputs as
"residence times" to clarify that we are only considering losses from transport processes. Two
steady-state residence times were defined here to analyze atmospheric dispersion processes, the
urban canopy column residence time ($\tau_{canopy}$), which is the column density of a species in the urban





canopy (i.e., the concentration integrated vertically from the ground to the urban canopy height)
divided by $L_{canopy}$ and the column export residence time ($\tau_{export}$) which is the downtown column
density of a species (i.e., the concentration integrated vertically from the ground to the top of the
model column) divided by the column export loss rate, $L_{export}$. When the model simulates idealized
cases and is run until steady state is achieved, these residence times are completely accurate. For
the model simulations of observed $SO_2$ during the ALPACA campaign, sources and sinks vary
with time such that the system is not at a true steady state. However, we assume it is "near steady
state" for the purposes of calculation of residence times, and as we will demonstrate, residence
times are short indicating that the system approaches steady state on similarly short timescales.

**4 Results**
**4.1 Conceptual model for pollution trapping in Fairbanks**
Figure 2 shows a conceptual model of the horizontal and vertical dispersion processes in
Fairbanks and was used to inform the setup of the PACT-1D simulations for Fairbanks. In Figure
2, the horizontal arrows representing wind show that there is little to no wind present near ground
level and wind increases with altitude, staying relatively slow in the urban canopy (due to friction
on ground roughness) and increasing to the top of the SBL. The vertical arrows in Figure 2
represent vertical transport based on the SBL parameterization described in the introduction, where
vertical exchange between layers increases from near zero at the ground level to a maximum at
roughly 20% of the SBL height, then decreases to near zero above the SBL.
A series of idealized model simulations with simplified boundary conditions that were
constant in time (*i.e.*, *h,* wind speed profile, emissions) were performed to observe approach to
steady state and verify model closure. These idealized model simulations based on the conceptual





model in Figure 2 enable a validation and a more intuitive understanding of the model scheme.
Figure 3 shows results from simulations that used a constant emission rate, time-invariant vertical
profile of wind speed and constant SBL height, $h$, as shown in Table 2. The $SO_2$ emissions rate is
$6 \times 10^{-9}$ moles $m^{-2}$ $s^{-1}$ $SO_2$, similar to the average wintertime $SO_2$ emissions in Fairbanks.

The case 1 simulation ($h$ = 25 m, 2 m $s^{-1}$ wind above SBL) achieves a 3 m AGL steady

state $SO_2$ mixing ratio of ~35 nanomole $mole^{-1}$ in about four hours (Figure 4). When the input SBL
height is doubled from 25 m in case 1 to 50 m in case 2, the peak value of $K_z$ roughly doubles,
accelerating vertical export from the urban canopy. The result of this doubling of the SBL height
is to cut the steady state urban canopy $SO_2$ mixing ratio roughly in half. When the wind speed aloft
is changed from 2 m $s^{-1}$ to 5 m $s^{-1}$ without changing the SBL height in case 2 and case 3, the steady
state $SO_2$ mixing ratio decreases further because there is a greater horizontal export in the polluted
layers from 15 m to 50 m. In the case 1 and case 2 simulations with a constant wind speed of 2 m
$s^{-1}$ aloft and an input SBL height $h \lesssim 50$ m, the height of the polluted layer is nearly the input SBL
height, yielding a sigmoid shaped profile (see Figure 3). In contrast, case 3 shows that higher winds
aloft remove pollution in upper layers more effectively and yields a more triangular shape with the
effective height of the polluted layer is somewhat shorter than the input SBL height, $h$. Case 4
shows that for a taller SBL height (100 m), even with low winds, the simulation gives a triangular
vertical profile.

Figure 4 shows a false-color plot of the vertical profiles of $SO_2$ over a 12-hour simulation

for case 2 (top panel) and the ratio of the vertical loss rate out of the urban canopy, $L_{canopy}$, and the
downtown column horizontal export loss rate, $L_{export}$, to the constant emissions rate for $SO_2$,
showing that the model achieves a steady state in about four hours, where $SO_2$ accumulates in the
surface layer until the emissions and the loss rate out of the SBL balance each other. Note that the





vertical exchange precedes the horizontal export because vertical transport into the windy region
from 15 to 50 m is required for pollution export. As expected, when steady state is achieved, the
horizontal export loss rate from the model and the upward loss rate from the urban canopy are both
equal to the emission rate. This simulation demonstrates that for realistic conditions in Fairbanks,
near steady state is achieved in a few hours, which indicates that use of the steady state residence
time to diagnose time-varying simulations is reasonable.

**398    4.2 Observations and modeling for the ALPACA campaign**

Figure 5, panels A through E, show time series of different measurements during the

ALPACA field campaign from January 15 to February 28, 2022. There were multiple times during
the ALPACA campaign that had low temperatures (panel A), persistent temperature inversions, as
indicated by a positive 23 m minus 3 m temperature differences ($dT_{[23m-3m]} > 0$) (panel B), small
wind speeds on the 23 m vertical scale (panel C), differences in $CO_2$ on the 23 m vertical scale
(panel D, with more $CO_2$ at 3 m than at 23 m) and accumulation of $SO_2$ at ground level (panel E).
During the persistent temperature inversion event that occurred during the "cold-polluted period"
(January 30[th] through February 4[nd]), $CO_2$ reached a relatively constant mixing ratio at 3 m AGL
by around midnight on January 31[st] and maintained similar levels until SBL breakup by winds
from aloft on February 4[th].

With ongoing emissions throughout the persistent inversion event, this near-steady

pollution behavior can only be explained by a pollution export from the SBL, as in the conceptual
model shown in Figure 2. At times when the SBL is above the higher altitude (23 m AGL), $CO_2$
reaches equivalent mixing ratios at both 3 m and 23 m AGL and conversely when the SBL is near
or below the top $CO_2$ sensor, mixing ratios are larger at 3 m and smaller at 23 m AGL. There are





even periods (e.g., on February 3$^{rd}$) when $CO_2$ at the 23 m altitude is near the regional background
of ~420 micromole mole$^{-1}$, indicating it is above the SBL, in the free troposphere and that the SBL
height must be below 23 m.
Figure 6 panels A and B show false-color time series of the model output $SO_2$ mixing ratio
and wind speed profiles, respectively. The model was not sensitive to changes in the time step for
dispersion processes, as there was no change in the slopes and $R$ values for either $SO_2$ or $CO_2$ when
the time step in the model (typically 5 seconds) was doubled to 10 seconds or decreased to 2.5
seconds. Figure 6 also shows a time series of the downtown column emissions in panel C and
model input SBL height in panel D. When the SBL height is small, pollution is trapped in a
similarly short and increasingly concentrated layer in Figure 6. At times when the SBL is near
neutral (off scale vertically in panel D), pollution mixing ratios were small throughout the vertical
column, as would be expected for fast vertical mixing and dilution implied by a tall SBL height.
The modeled concentration profile time series shown in Figure 6 panel A can be integrated
vertically between the LP-DOAS base altitude (17 m) and the height of the retroreflector on that
LP-DOAS path to give the model-predicted path-averaged $SO_2$ mixing ratio for that path.
Therefore, the four LP-DOAS paths, here denoted as P0 to P3 represent the average mixing ratio
from 17 m to 12 m, 73 m, 115 m, and 191 m AGL at the LP-DOAS base, respectively.
Figure 7 panels A through D show time series plots of the hourly path-averaged $SO_2$ from
PACT-1D, LP-DOAS field observations, and in-situ 3 m field observations (panel A only). In
Figure 7 panel A, the model path 0 (P0, 12 m to 17 m AGL) averaged $SO_2$, the LP-DOAS path 0
average $SO_2$ and the in-situ 3 m $SO_2$ all show good agreement in the time series, with mixing ratios
of 20 to 40 nanomole mole$^{-1}$ observed in the urban canopy layer when polluted. The trapping of
$SO_2$ near ground level in Figure 7 panel A occurs at times with surface-based temperature





inversions (low SBL height) and slow wind speeds within the urban canopy layer. In Figure 7
panel B, the timing of $SO_2$ peaks observed by the LP-DOAS on path 1 (P1, 17 m to 73 m AGL)
matches the peaks observed on path 0, in panel A, however the magnitude of $SO_2$ pollution is
smaller, only reaching up to 15 nanomole mole$^{-1}$ in panel B. In Figure 7 panels C and D, the
magnitude of $SO_2$ pollution observed continues to decrease with altitude and there are also some
$SO_2$ peaks observed by the LP-DOAS on paths 2 (P2, 17 m to 115 m AGL) and 3 (P3, 17 m to
191 m AGL) that are not present in the path-averages modeled by PACT-1D, particularly during
the persistent surface-based temperature inversion event during the "cold-polluted period" in
Figure 5. We hypothesize that these peaks on paths 2 and 3 are due to elevated power plant
emission plumes that are horizontally transported into the upper light paths. However, this 1-D
model is not suited for simulations of power plant emission dispersion, an inherently 3-D, process,
so later 3-D modeling would be required to analyze these infrequent spikes in the upper two LP-
DOAS path data.


**4.3 Correlation of modeled pollution to observations**

Figure 8 shows four modeled versus observed $SO_2$ correlation plots of data averaged at 3-

hour time resolution, which was used to diagnose the skill of the model in simulating observed
$SO_2$ mixing ratios. Linear fits are shown with either a constrained zero intercept or a free intercept.
Figure 8 panel A represents the overall agreement between LP-DOAS and model path-averages
across all paths, panel B represents the agreement of the model to in-situ $SO_2$ observations at 3 m
AGL and panels C and D represent the agreement between the model and LP-DOAS path-averages
in the first two paths (path 0 and path 1, respectively), which are the paths where a majority of the





pollution observed by the LP-DOAS is located. Correlation plots of the modeled path 2 and path 3
SO$_2$ versus the LP-DOAS are shown in Figure S3 in the Supplemental Materials. The PACT-1D
model showed good agreement with observations, as the *R* values for all DOAS path-averaged
data (Figure 8 panel A) and in the urban canopy (Figure 8 panels B and C) are all greater than 0.8.
The observed and modeled SO$_2$ mixing ratios on the upper two paths are usually small and the
correlation in Figure 8 panel A is dominated by data from path 0 and path 1. Figure 7 panel A
shows that the path-averaged path 0 SO$_2$ observed by the LP-DOAS has good agreement with the
in-situ SO$_2$ observed at 3 m AGL, with a zero-intercept linear correlation *slope* = 0.95 and *R* =
0.89 (see Figure S4), suggesting that SO$_2$ is well mixed within the ~15 m AGL urban canopy layer.

Figure 9 shows a time series of hourly averaged model and in-situ CO$_2$ at 23 m (middle

panel) and at 3 m (bottom panel) and the modeled and observed 23 m minus 3 m CO$_2$ difference,
d$CO_{2[23m\text{-}3m]}$. The model simulated observed CO$_2$ with good agreement, with a 3-hour average
free-intercept *slope* = 0.75 and *R* = 0.83 for the 3 m CO$_2$ data and a *slope* = 0.73 and *R* = 0.74 for
the 23 m CO$_2$ data (see Supplemental Materials Figure S5). The top panel of Figure 9 shows that
while the model does capture the timing of pollution trapping in the first 23 m, there are still many
times when the model overestimates CO$_2$ at 23 m.

**4.4 Model residence times during ALPACA**

For the duration of the ALPACA field campaign, the urban canopy residence times ($\tau_{canopy}$)

and column export residence times ($\tau_{export}$) for SO$_2$ were calculated from PACT-1D model outputs
(see Supplemental Materials Figure S6). Figure 10 shows the log-time distribution of the urban
canopy (left panel) and column export (right panel) SO$_2$ residence times. The urban canopy
residence times in the left panel of Figure 10 have a bimodal distribution, where the mode with





shorter residence times from 0.1 hours (6 minutes) to around 0.33 hours (20 minutes) occur during
near-neutral surface atmospheric stability conditions and the mode with longer residence times
from around 0.33 hours (20 minutes) to 2.4 hours occurs during stable atmospheric conditions.
During stable conditions with an SBL height, $h \lesssim 50$ m, data in the left panel of Figure 10
has a log-normal distribution with a median $\tau_{canopy} = 54$ minutes. The median urban canopy
residence time of the data overall was $\tau_{canopy} = 40$ minutes. The histogram in the right panel of
Figure 10 has log-normal distribution has overlapping modes indicating that the column export
residence time is less affected by changes in atmospheric stability, with a median $\tau_{export} = 1.8$ hours
overall and $\tau_{export} = 2.1$ hours during stable conditions. In Figure 11, the urban canopy and column
export $SO_2$ residence times are plotted against the modeled SBL height from Table 1. The urban
canopy residence time in the top panel in Figure 11 is anti-correlated with SBL height, increasing
with decreasing SBL height. Some of the column export residence time data in the bottom panel
in Figure 11 is anti-correlated with SBL height, but there is more spread in the data as short
residence times are sometimes observed at times with shallow SBL heights.
**5 Discussion**
**5.1 Relationship between steady state vertical profile shape and SBL height**
Figure 3 shows that the shape of the $SO_2$ vertical profile depends upon the SBL height and
wind profile. For shallow SBL heights and low winds aloft, the model simulates a sigmoidal shape
with the polluted layer extending to the SBL height, while for taller SBL height and higher winds
aloft, the $SO_2$ profile becomes a more triangular shape and the effective pollution layer height is
below the SBL height, as observed when comparing the Figure 3 cases 3 and 4 simulation to the
case 1 and case 2 simulations. This shows that the SBL height is a critical parameter for simulating





the vertical distribution of pollution in Fairbanks, but also that the wind speed aloft affects the
shape.
The conceptual model presented in Figure 2 assumed there is an urban canopy from zero
to 15 m AGL where wind speed is low. This canopy has roughness elements, buildings and trees,
throughout it, which convert small winds into turbulence, enhancing mixing in this layer, while
slowing horizontal wind speed. The observations demonstrate that this layer is mechanically mixed
because the LP-DOAS path 0 average (average measurement height of 15 m) was well-correlated
with the in-situ $SO_2$ measured at 3 m at the CTC site (zero-intercept *slope* = 0.95 and *R* = 0.89).
At times when the winds have effectively stagnated in the urban canopy layer, export of pollution
occurs by upward transport from the urban canopy to windier layers aloft, as shown in the idealized
simulation represented in Figure 4 and vertical canopy flux approaches the source flux before
winds aloft can export this pollution, as represented by the export flux. In this idealized situation,
steady state is achieved in a few hours.


**5.2 Relationship between pollution trapping and temperature inversion strength**
In Table 1, the SBL height decreases with increasing surface-based inversion strength and
does not exceed 50 m AGL unless the inversion weakens enough to allow the SBL to be near
neutral. When the atmosphere is near neutral, a large value of $h$ = 400 m is used to allow larger
amounts of vertical transport to occur in the boundary layer, which leads to the dilution of pollution
in the model column representing downtown Fairbanks. This process is responsible for the rapid
cleanout events that occur episodically throughout the winter (*e.g.*, see Figure 5).





At times when the SBL height is very shallow, $CO_2$ accumulates to large mixing ratios near

ground level and vertical differences in $CO_2$ are seen across the CTC building in Figure 6, as
vertical transport is limited to a shallow vertical extent within the SBL and winds above the SBL
can quickly dilute pollution to background mixing ratios aloft. There are few of these periods with
extremely large $dT_{[23m-3m]}$, such as the end of the "cold-polluted period" (February 3rd) and the
"warm-polluted period" at the end of ALPACA from February 23rd through 25th, and late on
January 23rd. During these times, the $CO_2$ sensor at 23 m AGL was nearly at regional background
$CO_2$, about 420 micromole mole$^{-1}$, as would be expected if it were in the free troposphere, above
the SBL height. These observations help to verify that the urban canopy in downtown Fairbanks
can be below 23 m at times when $dT_{[23m-3m]}$ is very large, as was used in the parameterization of $h$
versus $dT_{[23m-3m]}$ in Table 1. Past work also demonstrated significant trapping of $PM_{2.5}$ on the 20 m
vertical scale under strongly stable conditions in Fairbanks, further motivating the minimum value
of $h$ used in Table 1 (Cesler-Maloney et al., 2022).

When choosing the maximum SBL height in Table 1, we knew this height must exceed

23 m during weaker surface-based inversions, as there were times when $CO_2$ mixing ratios were
more equivalent at 3 m and 23 m during inversions in Figure 6. In Figure 7, the path averaged
mixing ratios of $SO_2$ during surface-based temperature inversions were largest in path 0 from 12 m
to 17 m, reaching a maximum of ~40 nanomole mol$^{-1}$, and were smaller in path 1 from 17 m to
73 m, reaching a maximum of ~15 nanomole mol$^{-1}$. These $SO_2$ observations show that during
inversions, the SBL height is greater than 17 m but doesn't reach 73 m, as there must be cleaner
air somewhere in path 1 to make the path averaged $SO_2$ mixing ratio less than what is observed in
path 0. Therefore, we chose a maximum SBL height of 50 m during weak surface-based inversion
conditions in Table 1. Between the very strong inversion the weak inversion cases, we mapped out





an inverse curve that was then interpolated to give the SBL height from the near-surface inversion
temperature difference. These choices of SBL heights in Table 1 were informed by observations
and proved to model observations with excellent agreement. Sensitivity studies were also carried
out and discussed in Section 5.4 to examine the result of choosing higher or lower values of the
SBL height, $h$.

**5.3 Skill of PACT-1D in modeling observed SO$_2$ vertical profiles**

The modeled SO$_2$ path averaged and 3 m SO$_2$ mixing ratios have excellent agreement with

observations when PACT-1D simulated the full ALPACA campaign period, as indicated by the $R$
values of 0.88 and 0.81 in Figure 8 panels A and B, respectively. The home heating emissions and
the wind speeds used to calculate the horizontal exchange coefficient representative of Fairbanks
based on the best available data from ADEC and EPA. The overlap between the SBL height and
the vertical wind speed profile is the most important factor controlling pollution trapping in
Fairbanks. When the SBL height is large enough, emissions are mixed more quickly through a
taller section of the vertical column, where they intersect with altitudes having larger horizontal
exchange. This vertical ventilation results in a more well-mixed column with small mixing ratios
of SO$_2$ at ground level, as observed both in the model and in the DOAS data at times when the
SBL is near neutral ($h > 50$ m) in Figure 6.

**5.4 Model sensitivity studies**

To explore the sensitivity of model results to input parameters, a series of simulations were

carried out, as described in Table 2, which varied parameters from the "Base" simulation described
above. The zero-intercept linear correlation slopes and $R$ values for the "Base" simulation and each





sensitivity study are reported in Table 3. The y-intercepts for free-intercept fits are shown in
Supplemental Materials, Table S1 and are typically just a few nanomole mole$^{-1}$. The path 1 to path
0 slope ratio (Table 3) was the metric used to gauge the model's success in simulating the observed
vertical shape of $SO_2$ above Fairbanks. In Table 3, decreasing the SBL height decreased the slope
ratio from the "Base" 1.00 to 0.59, while increasing the SBL height increased the P1/P0 slope ratio
to 1.47, showing that the $SO_2$ vertical gradients are highly sensitive to changes in SBL height.
Therefore, we find that the model parameterization's very shallow SBL heights, $h = 20$ to $50$ m
are required for a realistic simulation of the vertical shape of the polluted layer.

Changing the parameter for Fairbanks horizontal length, $L$, changes the magnitude of the

model-observation comparison slopes in Table 3, effectively increasing or decreasing the amount
of pollution measured near ground level. The value chosen in the "Base" simulation is 4 km, which
is the horizontal box edge dimension of the urban area being simulated (Figure 1). Changing the
Fairbanks model length, $L$, does not change the shape of the vertical profile of $SO_2$, as indicated
by the relative insensitivity of the P1/P0 slope ratio to changing $L$. This shows that the skill of the
model in simulating the correct vertical profile of pollution is not sensitive to changes in $L$; this
parameter only affects the amount of pollution trapped in the model. Changing the emissions rates
using a multiplier has a similar effect in the model as changing $L$, where increasing or decreasing
the emissions rates increases or decreases both path 0 and path 1 slopes but does not change the
P1/P0 slope ratio. Therefore, to the extent that the emissions are not known accurately, the effective
horizontal export rate cannot be uniquely determined. However, the ADEC's best estimate of
ground level $SO_2$ emissions, which are only from home heating sources, was used in the downtown
areas, and the transport length is realistic with regards to the physical size of downtown Fairbanks,
indicating the model performs well with Fairbanks emissions and size.



The constant near-neutral SBL height, $h_{neutral}$ = 400 m used when the atmosphere near the
surface is less stable, as indicated by times when $dT_{[23m-3m]}$ < 0.2 °C, was also studied. The model
was found to be nearly insensitive to $h_{neutral}$. This validates our choice of 400 m for $h_{neutral}$.
Although the friction velocity, u*, certainly varies over time, the simulation used a reasonable
fixed value, which was perturbed to be $u*$ = 0.6 m s$^{-1}$ or $u*$ = 0.25 m s$^{-1}$ in sensitivity runs Table 3.
This perturbation shows that varying the value of $u*$ had minimal effect on the slopes and $R$ values
and that the model results are fairly insensitive to the value of $u*$. Overall, the model is mostly
sensitive to the SBL height for the vertical shape and to the source (emissions) and sink (affected
by Fairbanks length) for the magnitude of pollution at ground level. Periods when the inversion
"breaks" are critical for cleaning out pollution and the timing of these events are well predicted by
weak temperature inversions and winds penetrating the urban canopy.

**5.5 Analysis of modeled steady state transport residence times**
The urban canopy column residence time, $\tau_{canopy}$, in the "Base" simulation is typically
below an hour, reducing the amount of chemical processing that can occur within the urban
canopy, where pollution accumulates to large amounts. While the median column export residence
time, $\tau_{export}$, is also relatively short for the downtown Fairbanks area, there are times when this
residence time exceeds 5 hours and even approaches 10 hours, suggesting that there may be times
when a reservoir of pollution aloft has more time for chemical processing to occur (see Supplement
Figure S6). If pollution is transported above the urban canopy layer when the SBL height is taller,
it could become trapped in a reservoir aloft if the SBL height then decreases to a shorter altitude.
If this "lofted reservoir" of pollution is formed after a decrease in SBL height, then the residence
time of pollution in this lofted reservoir will depend upon the horizontal wind speed. If the



horizontal wind speed is large within the lofted reservoir, it will be quickly diluted by background
air from outside of Fairbanks and will not downwash processed pollution locally. However, if the
horizontal wind speed is slow within the lofted reservoir, there may be time for changes in
chemistry to occur within the reservoir and downwash of processed pollution into the urban area
could be possible.

**6 Conclusion**

A version of the PACT-1D model that includes horizontal advection and having vertical
dispersion controlled by an eddy diffusivity ($K_z$) profile based on literature SBL parameterization
and driven by a SBL height parameterization based on the near surface temperature inversion
strength, was used to simulate $SO_2$ as a tracer for dispersion. The model was successful in
simulating observed in-situ 3 m $SO_2$ and LP-DOAS path averaged $SO_2$ from 12 m to 191 m with
excellent agreement, with a modeled 3 m versus in-situ 3 m $SO_2$ correlation coefficient $R = 0.81$
and a model path average versus LP-DOAS path average $SO_2$ correlation coefficient $R = 0.88$ for
all paths.
Idealized steady state simulations validated the model's response to input parameters and
yielded vertical profiles and ground level steady state mixing ratios of $SO_2$ similar to what was
measured by LP-DOAS in the field. When the SBL height is shorter, the height of the polluted
layer is equivalent to the SBL height, as pollution can vertically mix throughout the shorter SBL
height with less horizontal loss by wind, yielding a sigmoid shaped profile. When the SBL height
is larger or winds aloft are stronger, winds aloft can dilute pollution that mixes vertically aloft,
such that the height of the polluted layer is shorter than the SBL height and the profile is more
triangular in shape. The SBL height derived from near-surface temperature difference observations





in downtown Fairbanks ($dT_{[23m\text{-}3m]}$) is very shallow when the atmosphere near the ground is stable,
never exceeding $h = 50$ m AGL and becoming as low as $h = 20$ m AGL for extreme inversion
strength. When the atmosphere is near-neutral in the region near ground level (when $dT_{[23m\text{-}3m]} <$
0.2 °C), the model is insensitive to SBL height as long as the SBL height is reasonably large, such
as the height used in the model at these times, $h_{\text{neutral}} = 400$ m.
The model successfully captured the vertical profile of $SO_2$ observations in downtown
Fairbanks. When comparing the shortest LP-DOAS path averaged $SO_2$ mixing ratio versus the in-
situ 3 m $SO_2$ measured at the CTC site, the zero-intercept linear regression analysis had a *slope* =
0.94 and $R = 0.89$, indicating that pollution is fairly well mixed within a ~15 m urban canopy layer
on a three-hour timescale. A series of sensitivity tests were performed to evaluate the model's
dependence on input parameters. The model's vertical profile was only sensitive to the SBL height,
and generally the surface mixing ratio responded linearly to variations in emissions and the length
of Fairbanks, which controls the horizontal exchange rate. The model is fairly insensitive to
variations in the neutral BL height and the value of the friction velocity.
The residence time of air in the urban canopy is usually shorter than a few hours, with a
median residence time of 40 minutes overall and 54 minutes when stable, and a median column
export residence time of 1.8 hours overall and 2.1 hours when stable, limiting time for chemical
processing to occur near the ground. Chemical processing may occur within a lofted reservoir of
pollution, which can either be removed from the Fairbanks area by horizontal advection, mixed
back down into the urban canopy given enough time, or possibly re-circulated back into the urban
canopy. However, the model achieves excellent results ($R = 0.88$) without considering the
recycling of pollution or lofted sources such as powerplants, indicating that these effects are not
dominant.




**Code / data availability**

Final data from the study is available on the Arcticdata.io ALPACA data portal

(https://arcticdata.io/catalog/portals/ALPACA). From this repository, we include the gas
composition, temperature, and wind data from the CTC site, doi:10.18739/A27D2Q87W. The
model code and input files will be uploaded to the portal upon acceptance of the manuscript and
are provided as a link to reviewers for the purpose of peer review.

**Author contribution:**

This work was originally authored by M.C.-M. with substantial contributions from W.S. All
authors contributed to revisions and editing of the final manuscript. The work is based upon the
PACT-1D model, which was developed by J.T. and J.S., and modified for this application by M.C.-
M., W.S., J.K., and J.S. Pollution emissions were provided by D.H. and resampled by J.K. Input
and validation data was measured and provided by J.S., M.C.-M., S.C., W.S, and T.R. Project
conceptualization and funding for the field study was obtained by W.S., J.S., and the ALPACA
science team.

**Competing interests:**

The authors declare that they have no conflict of interest.

**Acknowledgements**

We thank the entire ALPACA science team of researchers for designing the experiment,

acquiring funding, making measurements, and ongoing analysis of the results. The ALPACA



project is organized as a part of the International Global Atmospheric Chemistry (IGAC) project
under the Air Pollution in the Arctic: Climate, Environment and Societies (PACES) initiative with
support from the International Arctic Science Committee (IASC), the National Science Foundation
(NSF), and the National Oceanic and Atmospheric Administration (NOAA). We thank University
of Alaska Fairbanks and the Geophysical Institute for logistical support, and we thank Fairbanks
for welcoming and engaging with this research. W.R.S. and M.C.-M. acknowledge support from
NSF grants NNA-1927750 and AGS-2109134. J.S., J.K., and S.C. acknowledge funding support
from NSF grants 1927936 and 2109240. J.L.T. is funded by the European Union's Horizon 2020
research and innovation programme under grant agreement No. 101003826 via project CRiceS
(Climate Relevant interactions and feedbacks: the key role of sea ice and Snow in the polar and
global climate system).  T.R. acknowledges support from the Agence National de Recherche
(ANR) CASPA (Climate-relevant Aerosol Sources and Processes in the Arctic) project (grant no.
ANR-21-CE01-0017), the Institut polaire français Paul-Émile Victor (IPEV) (grant no. 1215) and
Labex Voltaire (grant no. ANR-10-LABX-100-01). We thank Robert Gilliam from the U.S.
Environmental Protection Agency for providing the WRF wind speed data. Disclaimer: The views
expressed in this article are those of the authors and do not necessarily represent the views or
policies of the U.S. Environmental Protection Agency.

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





**Tables**
**Table 1:** Stable boundary layer height, $h$, parametrization as a function of near-surface stability.
These values are for the "Base" simulation; sensitivity studies vary either the stable ($dT_{[23m\text{-}3m]} =>$
0.2 °C or near-neutral ($dT_{[23m\text{-}3m]} < 0.2$°C) boundary layer height.

| $dT_{[23m\text{-}3m]}$ / °C | $h$ / m |
|---|---|
| <0.2 | 400 |
| 0.2 | 50 |
| 2 | 30 |
| 5 | 25 |
| 10 | 20 |






**Table 2:** Cases 1 through 4 used SBL parameterization (Equation EQ1) to calculate a vertical
profile of $K_z$ from a constant SBL height and a linear wind profile, with no wind in an urban canopy
from zero to 15 m and a linear increase from 15 m to a constant geostrophic wind above the SBL.
A constant $SO_2$ emissions rate of $6 \times 10^{-9}$ moles m$^{-2}$ s$^{-2}$ was used in each case.

| Variable | Case 1 | Case 2 | Case 3 | Case 4 |
|---|---|---|---|---|
| **SBL height** | $h = 25$ m | $h = 50$ m | $h = 50$ m | $h = 100$ m |
| **Wind speed** | 0 m s$^{-1}$ in the first 15 m to a constant 2 m s$^{-1}$ above SBL | 0 m s$^{-1}$ in the first 15 m to a constant 2 m s$^{-1}$ above SBL | 0 m s$^{-1}$ in the first 15 m to a constant 5 m s$^{-1}$ above SBL | 0 m s$^{-1}$ in the first 15 m to a constant 2 m s$^{-1}$ above SBL |







**Table 3:** Results of sensitivity studies for the full ALPACA period. Zero-intercept linear slopes
and intercepts are listed for 3-hour path-averaged modeled $SO_2$ versus LP-DOAS $SO_2$. The "Base"
simulation was set with $u* = 0.40$ m s$^{-1}$ and $L = 4$ km.

| Variation | LP-DOAS R | LP-DOAS slope | 3 m in-situ R | 3 m in-situ slope | Slope ratio P1/P0 |
|---|---|---|---|---|---|
| Base | 0.88 | 1.10 | 0.81 | 1.28 | 1.00 |
| $h = \tfrac{2}{3} h_{\text{base}}$ | 0.87 | 1.49 | 0.82 | 1.87 | 0.59 |
| $h = 1.5\, h_{\text{base}}$ | 0.86 | 0.84 | 0.80 | 0.91 | 1.47 |
| $L = \tfrac{2}{3} L_{\text{base}}$ | 0.88 | 0.82 | 0.80 | 0.99 | 0.95 |
| $L = 1.5\, L_{\text{base}}$ | 0.87 | 1.49 | 0.82 | 1.67 | 1.06 |
| $h_{\text{neutral}} = \tfrac{2}{3}$ base | 0.88 | 1.12 | 0.81 | 1.30 | 1.03 |
| $h_{\text{neutral}} = 1.5$ base | 0.87 | 1.09 | 0.81 | 1.27 | 0.99 |
| $u* = 0.25$ m s$^{-1}$ | 0.88 | 1.19 | 0.82 | 1.46 | 0.92 |
| $u* = 0.60$ m s$^{-1}$ | 0.87 | 1.04 | 0.80 | 1.16 | 1.07 |
| $E = \tfrac{2}{3} E_{\text{base}}$ | 0.88 | 0.75 | 0.81 | 0.87 | 1.02 |
| $E = 1.5\, E_{\text{base}}$ | 0.88 | 1.64 | 0.81 | 1.91 | 0.99 |




**Figures**

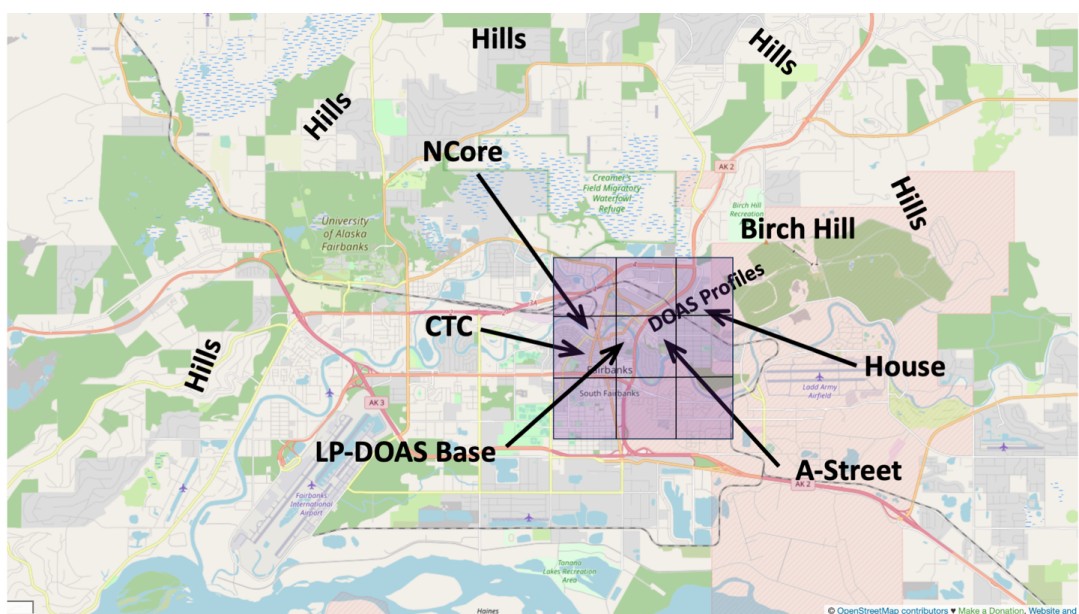


**Figure 1:** Map of measurement locations in downtown Fairbanks. The purple box represents the
area of the nine (3x3) grid cells of the ADEC emissions and WRF-modeled meteorological
variables used in PACT-1D. The WRF meteorological variables were taken from the center grid
box of the 3x3 grid. Base map was produced by OpenStreetMaps.org using data available under
the Open Database License see https://www.openstreetmap.org/copyright.



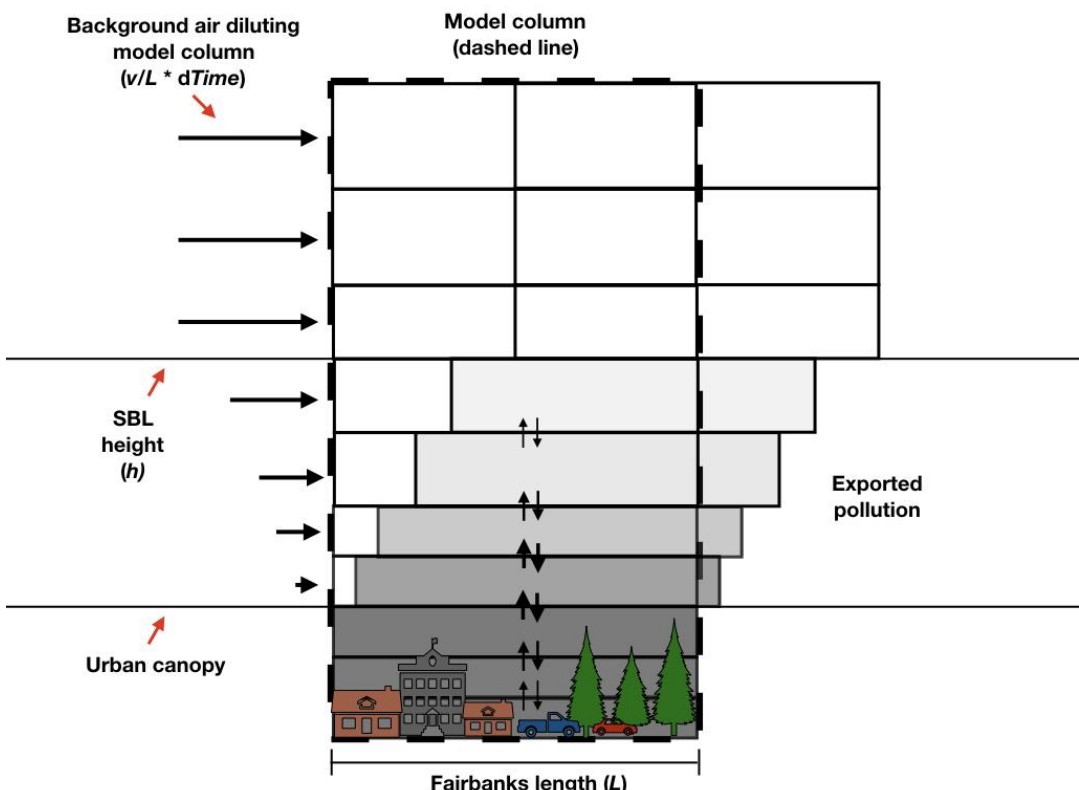

**Figure 2:** Conceptual model of emissions and dispersion processes used to describe Fairbanks, Alaska within the Platform for Atmospheric Chemistry and Transport one-dimensional model PACT-1D. Emissions occur within the urban canopy where there is no wind to export pollution. Vertical dispersion transports pollution into layers within the SBL where winds bring in clean background air to dilute pollution in the model column and export pollution out of Fairbanks. Vertical dispersion goes to zero both at the ground and at the top of the SBL such that no pollution reaches above the SBL, where there are fast moving, geostrophic winds aloft.



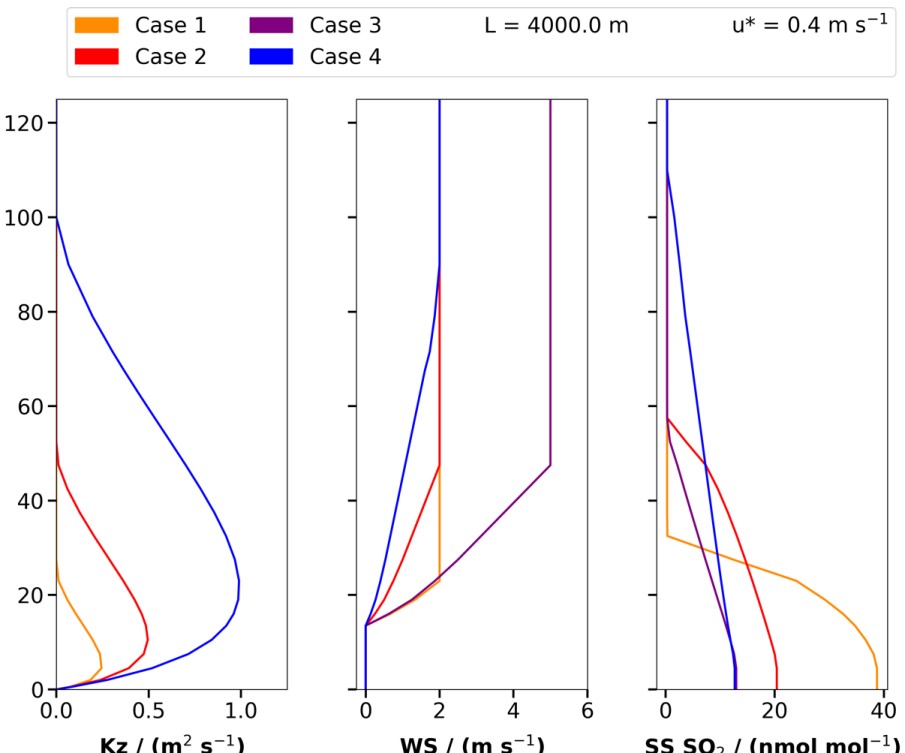

**Figure 3:** Vertical profiles of inputs and simulated $SO_2$ at steady state for various scenarios using downtown column emissions of $6\times10^{-9}$ moles $m^{-2}$ $s^{-1}$ $SO_2$. cases 1 though 4 use the SBL $K_z$ parameterizations from Equation EQ1 with different stable boundary layer heights, $h$, which are case 1: $h = 23$ m, cases 2 and 3: $h = 50$ m, and case 4: $h = 100$ m. Cases 1 through 4 use a wind model with zero winds in the urban canopy from zero to 15 m AGL, then a linear gradient increasing to achieve a free-atmosphere wind speed at the top of the stable boundary layer. The winds above the SBL are cases 1, 2, and 4: 2 m $s^{-1}$ and case 3: 5 m $s^{-1}$. The resulting modeled $SO_2$ profiles for each case are also plotted in panel D.



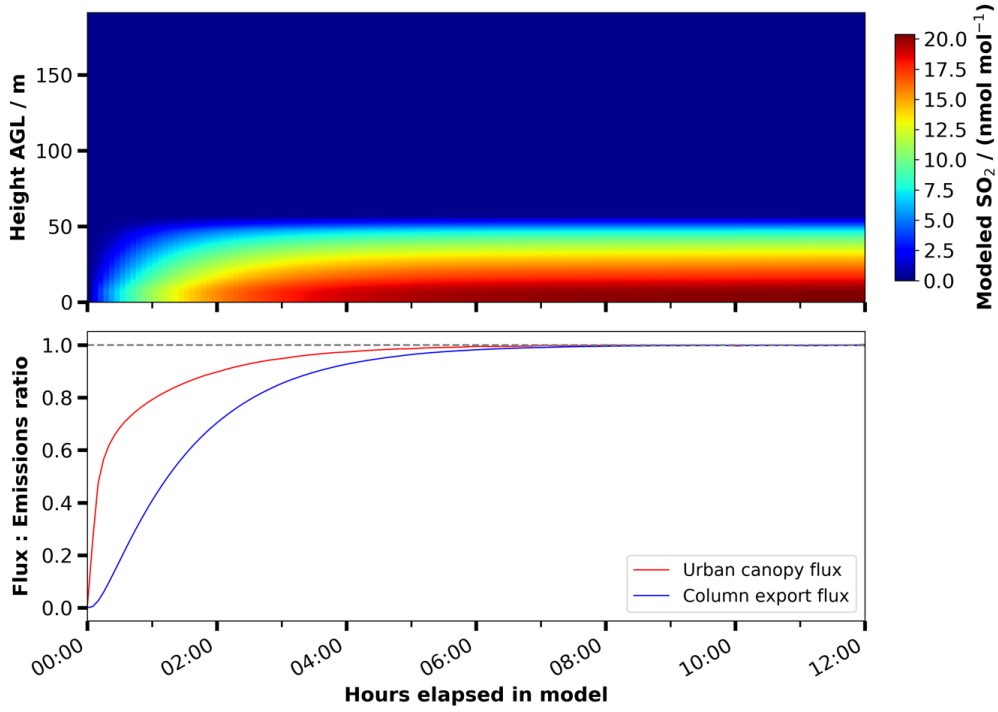

**Figure 4:** An altitude / time "curtain plot" showing $SO_2$ mixing ratio represented by colors for the case 2 steady state simulation with a constant 50 m stable boundary layer height. The ratio of the loss rate out of the urban canopy by vertical transport and the export loss rate out of the column by horizontal exchange to the downtown column $SO_2$ emissions are shown as time series on the bottom panel.



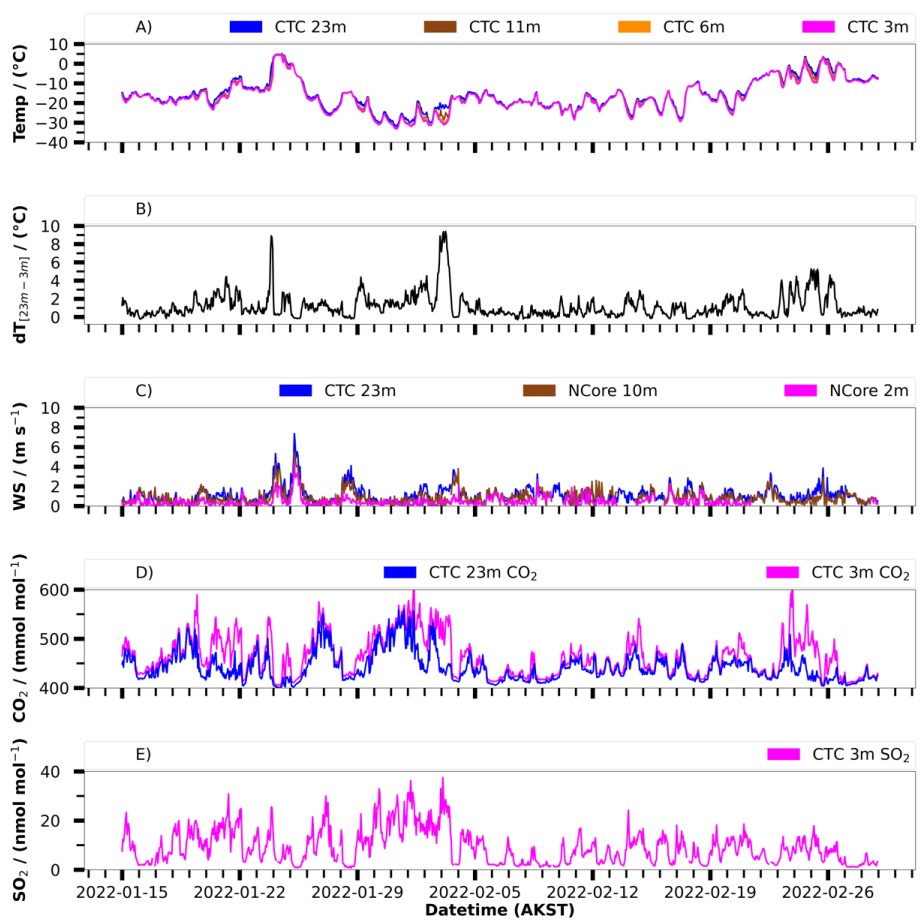

**Figure 5:** Time series of hourly averaged in-situ field measurements from the UAF CTC site during the ALPACA field campaign in January and February of 2022. The $CO_2$ in panel D is in micromole mole$^{-1}$ (mmol mol$^{-1}$ is an abbreviation for micromole mole$^{-1}$).



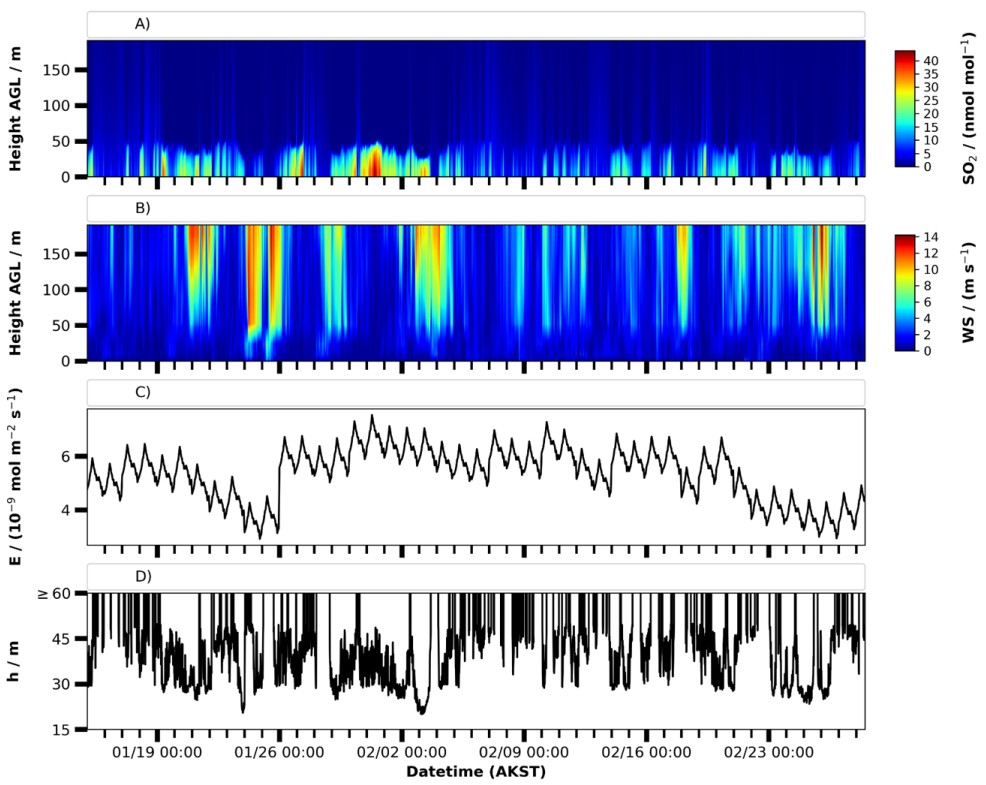

885

**Figure 6:** Panels A and B are altitude / time "curtain plots" showing model parameters and results

by colors for the ALPACA "Base" simulation. Panel A is modeled $SO_2$ and panel B is wind

speed, $v$. Panel C is a line plot of the $SO_2$ downtown column emissions rate, $E$, and panel D is a

line plot of the stable boundary layer height, $h$. Note that $h$ increases off scale to a boundary layer

height of 400 m AGL when the atmosphere is near neutral.



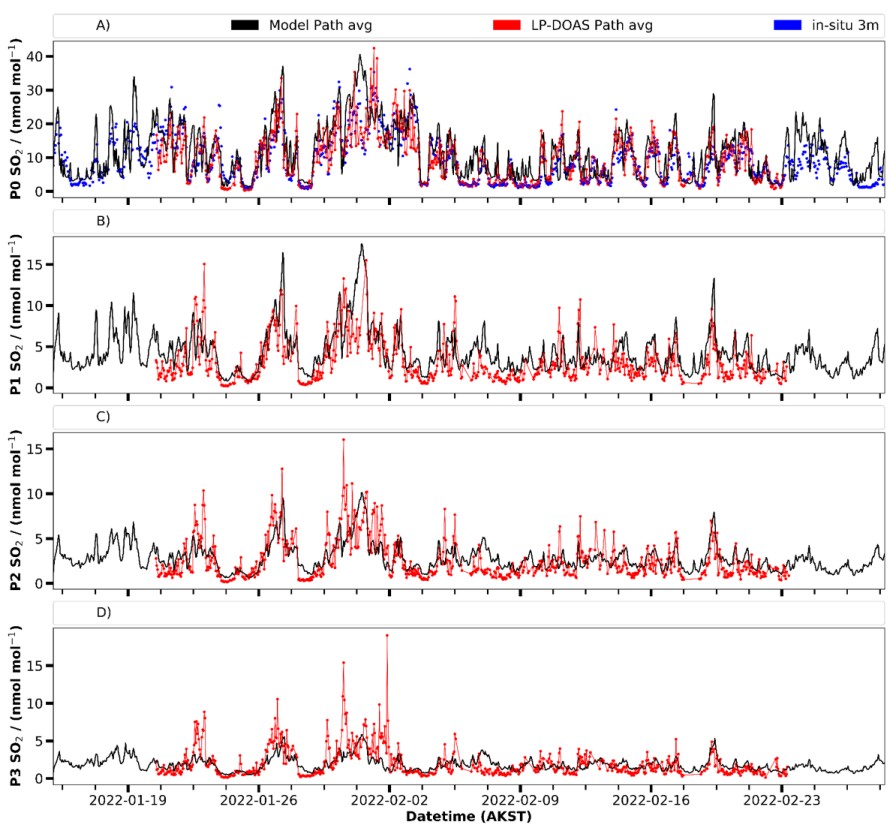

891

**Figure 7:** Time series of hourly path-averaged $SO_2$ from PACT-1D (black line), LP-DOAS field

observations (red dots) and in-situ 3 m field observations (blue dots in panel A). Model 3 m data

was not included in panel A as the correlation between model 3 m $SO_2$ and in-situ 3 m $SO_2$ was

good (zero-intercept *slope* = 1.28 and *R* = 0.81 in Table 1). Panels B through D show the hourly

path-averaged LP-DOAS observations and model results for path 1 through path 3, respectively.



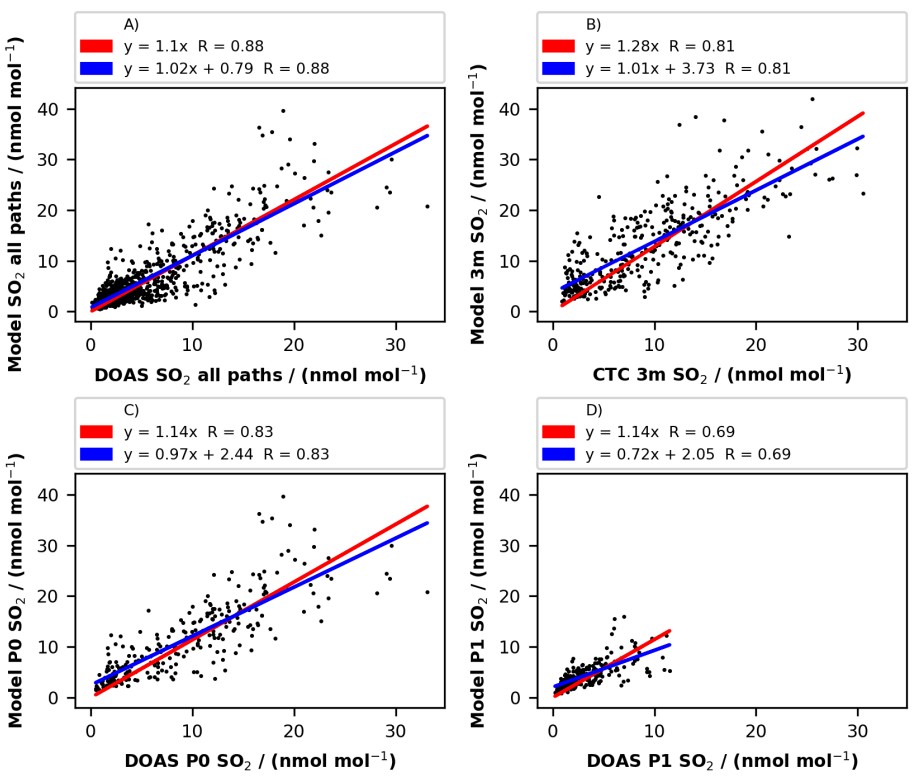

897

**Figure 8:** Correlation plots of 3-hour averaged model versus observed SO₂. Panel A shows the model path averaged SO₂ mixing ratio versus the path averaged SO₂ mixing ratio observed by the LP-DOAS for all four DOAS paths from 12 m to 191 m AGL. Panel B shows the modeled SO₂ in the 3 m to 6 m layer in PACT-1D versus the in-situ 3 m SO₂ observed at the CTC site. Panel C shows the model path 0 SO₂ versus the LP-DOAS path 0 SO₂ (path 0 being the average from 12 m to 17 m AGL) and panel D shows the model path 1 SO₂ versus the LP-DOAS path 1 SO₂ (path 1 being the average from 17 m to 73 m AGL). The red lines represent the zero-intercept linear correlation equations and the blue lines represent the free-intercept linear correlation equations, with coefficients for each equation shown in the legend on each panel.



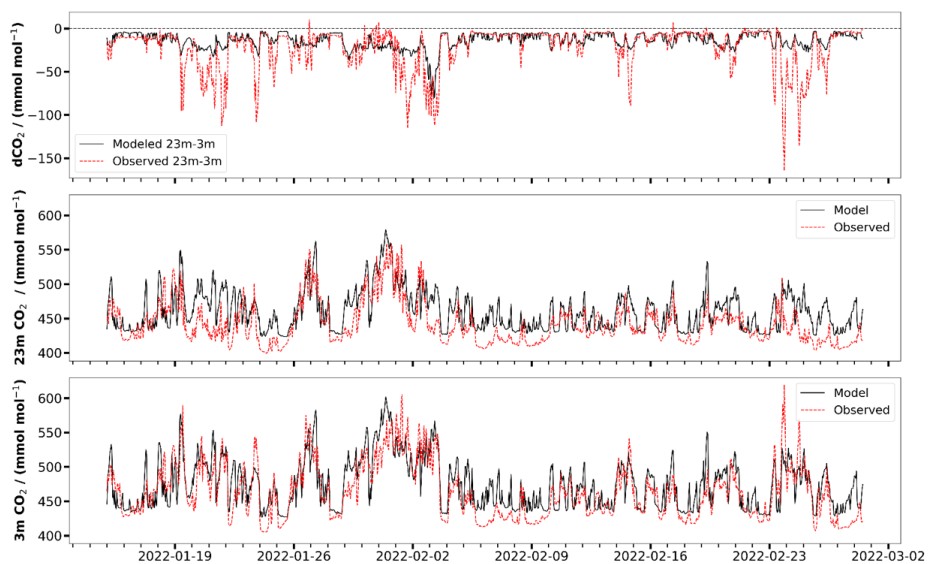

907

**Figure 9:** Time series of hourly averaged modeled and in-situ $CO_2$ measurements at 3 m (bottom

panel) and 23 m (middle panel) at the CTC site in downtown Fairbanks. The top panel shows the

modeled and observed 23 m minus 3 m $CO_2$ differences, $dCO_{2[23\ m-3\ m]}$, in micromole mole$^{-1}$

(mmol mol$^{-1}$ is an abbreviation for micromole mole$^{-1}$).

912

913



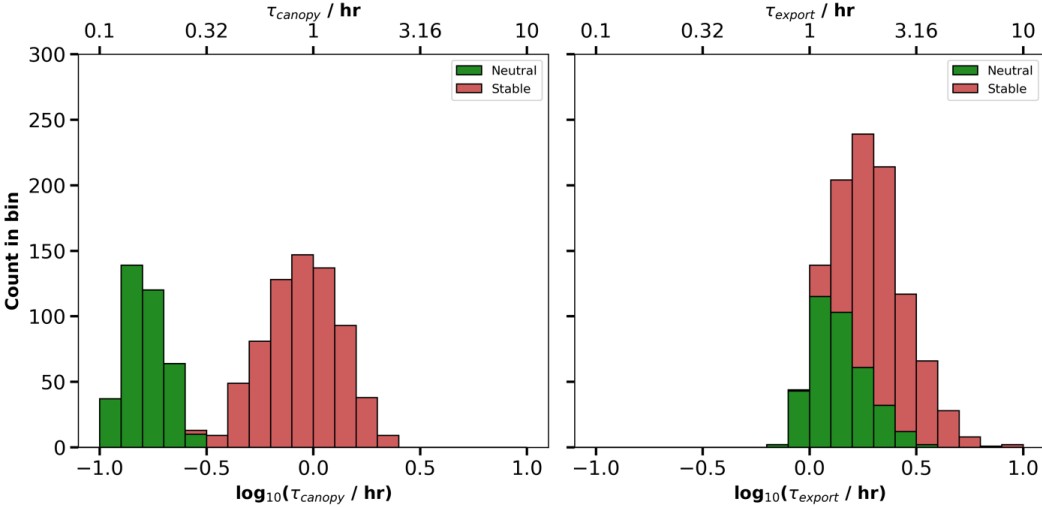

**Figure 10:** Histograms showing the ALPACA campaign simulated residence time distributions. The left panel shows the steady state urban canopy residence time and the right panel shows the steady-state column export residence time. Note that the histograms are done using logarithmic bins for the residence time on the bottom x-axis and the residence time in hours on the top x-axis.





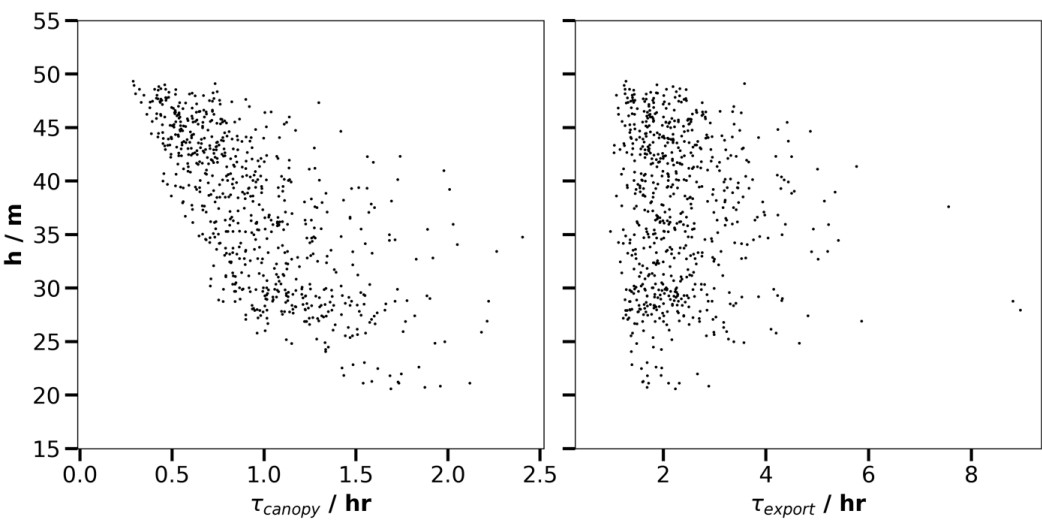

921

**Figure 11:** Steady state calculated residence times for the full ALPACA simulation. The left panel

shows the urban canopy residence time vs the model stable boundary layer height, *h*, and the right

panel shows the column export residence time vs *h*.