# Peer review of "Shallow boundary layer heights controlled by the surface-based temperature inversion"

_EGUsphere, 2023_

## Author Comment (AC2)

Reply to the Editor and both reviewers:

The main aim of this paper is to understand the dispersion of chemical pollutants within stable boundary layers. This information is needed to quantify chemical concentrations of species mixing in this environment and to predict chemical kinetic rates controlling pollution reactions. There is no paper in the literature that successfully predicts the vertical distribution of surface-emitted pollutant gases under temperature inversion conditions as extreme as were present during the ALPACA field study. This paper does not aim to make a new contribution to the field of boundary layer meteorology. Instead, we use established empirical boundary layer turbulence parameterizations to understand dispersion of pollutant gases during pollution episodes.

It is clear that the first reviewer viewed this manuscript as a boundary layer meteorology paper; therefore, the reviewer critiques that there is not new information for the field of boundary layer meteorology. Due to this misinterpretation of the manuscript, the reviewer has provided critiques that are orthogonal to the main aims. In addition, the comments on 1-D modeling indicate that the reviewer has not understood the published PACT-1D model papers or the model code and input/output that was provided. The first reviewer seeks more discussion of stable boundary layer theories and their predicted heights, but there is not a comprehensive theory for the boundary layer structure in these extremely stable situations and this paper is not an attempt to develop these.

The polluted layers like those affecting a small isolated city (e.g., Fairbanks, Alaska) are formed on a shorter timescale than the regional heat fluxes and synoptic-scale meteorological patterns that affect the temperature profile. Therefore, the height of stable polluted layers will typically be more shallow than those of temperature, which improved meteorological methods may seek to predict. Successfully quantifying the height of stable pollution layers and the vertical distribution of gases and particles is critical for the study of atmospheric chemistry, but their accurate representation has been a major challenge of prior atmospheric chemistry models. Here, we demonstrate that a simple empirical parameterization for the polluted boundary layer height as a function of the local surface temperature gradient performs excellently for the purposes of reproducing the surface-emitted pollution's vertical distribution near the surface. Neither reviewer mentions or critiques the success of this novel approach. The second reviewer's comments are not substantive, lack citations to literature, and provide only qualitative criticism.

For these reasons, it is clear that the reviews have not fully considered this manuscript. Therefore, we request a third review by an atmospheric chemist who can speak to the importance of this work for understanding coupled atmospheric chemistry and vertical mixing processes within stable boundary layers.

Reply to Reviewer 1:

It appears that the reviewer has misunderstood the topic of this manuscript and that the reviewer took this to be a boundary layer meteorology manuscript, which it was not. We did not

submit it to a dedicated boundary layer meteorology journal. We intended the audience to be interested in Atmospheric Chemistry, and the goal is to understand the dispersion of gases and particles (two identified topics of ACP), which is necessary to understand chemistry in stable boundary layers.

A critical aspect of this manuscript is the addition of horizontal export to an established 1-D Chemical/transport model. This model has been described in two recent publications (Tuite et al., 2021 and Ahmed et al., 2022), and for atmospheric chemistry purposes, was written to complement other 1-D models (e.g., MISTRA) to focus on increased understanding of vertically varying chemistry, particularly nocturnal chemistry. Unlike MISTRA, PACT-1D does not simulate boundary layer meteorology online. It takes Kz, temperature, pressure profiles from another model, observations, or calculation to constrain vertical mixing processes while calculating atmospheric chemistry (emissions, vertical / horizontal mixing and chemical transformations) online. In the two prior publications describing the model, the horizontal extent of the region in question was large and chemical lifetimes of species of interest were shorter, allowing the model to ignore horizontal export from the domain. As described in the introduction, the polluted region of Fairbanks is small enough that we need to consider horizontal export. Because of the strong temperature inversions (along with roughness of buildings and trees), air at ground level often decouples from layers aloft, leading to wind shear, which is taken into account in the horizontal export module. The results are quite novel in a number of aspects:

a) Observations of tracer dispersion clearly shows that air in contact with the ground (our definition of the boundary layer), can be as short as 20m AGL, and is typically below 50m during strong temperature inversions. We show that a simple parameterization of SBL height and vertical dispersion as a function of the near-surface temperature gradient is extremely successful.

b) We can successfully model not only the surface mixing ratio of the pollution tracer $SO_2$, but also its vertical profile with very high skill. Most studies only validate tracer mixing ratios at the ground, while our study has continual profiling for a five week period. If the parameterized vertical dispersion were wrong, the profile would be in error, while if the horizontal export were not included or were wrong, the amount of tracer would either grow unrealistically (if there were no export from the column) or be too dilute. Therefore, validating the model with tracer vertical profiles is novel and demonstrates the vertical and horizontal dispersion parameterizations are working exceptionally well.

c) The 1-D model predicts the residence time of pollution in this airshed and how the distribution of residence times varies as a function of strong versus weak temperature inversions. This residence time limits the time available for chemical transformations.

Below, we show the text of the reviewer in blue italics and our reply in black:

*-- Reviewer 1 --*

*Summary:*

*This study uses observations from the ALPACA-2022 field campaign in Fairbanks (Alaska) to understand the evolution of the SO2 concentrations in stable boundary layers over the city. The dynamics of the atmosphere and the vertical dispersion of the SO2 is modelled in the PACT-1D single-column model to learn about what are the controlling factors of the SO2 profiles in time and height. Fairbanks is subject often to very stable boundary layers, which trigger high concentrations and health issues. I have several concerns with the study in terms of novelty, physical consistency, reproducibility, focus and organization.*

*Recommendation: reject*

*Major remarks:*

*Research focus: my first concern is that the manuscript does not contain a well-defined research question. The introduction section is very long, while it does not include a proper review of SBL work done in the last 20 years (see later on about novelty), but it does not end up with a clear knowledge gap and a research question (or hypothesis) that will be answered (confirmed or refuted) in this study. The final stage of the Introduction announces what is going to be done as activities, but it lacks a constrained research question. Hence the focus needs to be sharpened. That is why the paper is also relatively long.*

As stated above, we take the criticism that the introduction should be improved, although adding more information about boundary layer meteorological theories, as requested, will make it a bit longer.  We agree to make the introduction clearer in the goals of the manuscript, describing how the model is useful for atmospheric chemistry.  The purpose of this study was to examine how strongly inverted conditions affected the mixing of a surface-released tracer.  We took the boundary layer to be the AMS Glossary of Meteorology definition: "The bottom layer of the troposphere that is in contact with the surface of the earth."  We used surface-emitted tracers ($SO_2$ and $CO_2$) to determine contact with the surface and measured the height to which the tracers mix by remote sensing or in-situ observation.  To try to keep the introduction shorter, we did not discuss that there is an interplay between the timescale of formation of a boundary layer and/or export of the surface tracer and the height through which the tracer mixes.  As discussed in Anderson and Neff (2008), the boundary layer with respect to temperature may differ from that of a surface-influenced tracer.  Anderson and Neff (2008) say, "Thus, the actual boundary layer may extend through only a small fraction of the depth deduced from the temperature profile, especially under stable conditions." In the case of the present study, as discussed in the manuscript, we use a tracer being released in a small geographical area (urban Fairbanks, represented by a footprint of 4km x 4km), so export of the pollution horizontally, which is shown to be on the order of a few hours, limits the timescale for "contact with the surface".  The temperature profile is affected by surface heat fluxes that may have happened on much longer timescales, and often more importantly by meteorological processes such as warm air advection.  To address the reviewer's question, a clearer statement of the research question is: Can we model the vertical distribution of a surface-emitted tracer over a five-week field study

including periods of extremely stable meteorological conditions alternating with periods of reduced stability? The answer is emphatically yes, and a novel aspect of this answer is that only the near surface temperature gradient and wind speed profile are needed to get these excellent results.

*Novelty: I find the study not novel. In my view the observations from the field campaign are new, and can offer insights. But the modelling work is building on studies done in the 1970s and 1980s, while much more insights have been developed later on. The K(z) parameterization that is being used has been developed for SBLs that are driven with relatively high wind speeds where the turbulence is well maintained, while in reality this is not the case, especially for shallow SBLs of ~50m or shallower that the authors study here. It is well know that the SBL has two regimes (e.g. Mahrt, 1998; Vogelezang and Holtslag, 1996), i.e. one (weakly stable boundary layer) where the turbulence is well-behaved and Monin-Obukhov scaling applies, and the very stable boundary layer where scaling does not work, and where other processes take over, like waves, longwave radiation divergence and so on. Alternatively a series of papers by Zilitinkevich categorized the SBLs in truly neutral SBLs, conventionally neutral SBLs, and long-lived SBLs. For each of them a diffusion transfer coefficient formulations have been built. The same holds for SBL height formulations. E.g. Steeneveld et al. (2007) developed an SBL height formulation that clearly distinguishes 2 regimes. None of this kind of archetyping is present in the submitted paper at hand.*

This comment makes it particularly clear that the reviewer misunderstands what is being done in this manuscript. The reviewer gives citations that discuss the boundary layer structures based upon micro-meteorological parameters and as diagnosed by methods other than tracer dispersion. These references are clearly important to boundary layer meteorologists and help to diagnose the boundary layer heights formed for temperature or humidity over long timescales, but as discussed earlier, the polluted boundary layer height is formed on a shorter timescale and differs from long-lived temperature structures. As the reviewer acknowledges, the observation of SBL height <50m and as low as 20m, observed through trace gas profiles, is unique.

In this manuscript, we show that a highly parameterized description of mixing is sufficient to give excellent agreement (R=0.88) with observed vertical profiles of a surface-emitted tracer. The fact that more complex boundary-layer meteorology theories exist does not mean that these empirical parameterizations don't work -- we show that the empirical parameterization in fact works very well. We did not say that there wasn't more complexity to the mixing or that turbulence was well behaved. In fact, we say "On short time and length scales, this mixing is highly complex, including intermittency and often showing oscillations (Mahrt, 1981)".

Note that our 1-D model requires an eddy diffusivity (Kz) profile, which we parameterize as a function of near-surface temperature gradient. Only the diffusivity profile is required by the 1-D model, not any of the parameters used in generating it (e.g., u*, L, the Obukhov length) are used in the modeling subsequent to the generation of this Kz profile. Effectively, these are empirical parameters used to generate the Kz profile. We make no claim that this old empirical continuum theory represents the actual mechanism of mixing. However, we find that this theory

does give a Kz profile that very effectively describes average mixing processes as shown by the excellent agreement between modeled and vertically resolved tracer concentrations.  Literature cited also shows that different approaches to the problem come to similar eddy diffusivity profiles.  For example, in Beare et al. (2006), a large eddy simulation (LES) model is used to study turbulence, and it is found that the eddy diffusivity profile is in agreement with the old empirical theory we use here.  Overall, we show excellent performance of an empirical model for diffusivity and stay away from interpreting the parameters.  We also show that the model is not even very sensitive to the assumed parameters. Following the reviewer's remarks we will clarify the manuscript introduction to further place the model in context, and to avoid misunderstandings between the meteorological and atmospheric chemistry research communities on this topic.

*Physical consistency. I am concerned about the physical consistency of the meteorological forcings in the 1D model. If I understand it well the 1D model has a prescribed PBL depth (from the observed T profile in the mast) and friction velocity (0.4 m/s) and a non-evolving wind speed profile. The paper is quiet about whether potential temperature and humidity is prognostic or not, neither whether longwave radiation divergence is accounted for. Obviously the SBL depth a result of the evolving wind speed, surface friction velocity, temperature profile etc. Earlier studies showed the equilibrium SBL depth is roughly 700 times the friction velocity (see Vogelezang and Holtslag paper), or alternative expressions where h ~ u\*. Here if h =700 \* 0.4 means the eq PBL depth would be 280 m, which is much deeper than the shallow PBLs the paper discusses. So it appears that for many of the simulations one is in a unphysical range/domain for geowind, friction velocity and SBL height. If this is true the results are not meaningful.*

The reviewer misunderstands our single column model.  Our model's meteorology is constrained by observations and is not "free running".  The purpose of the model is to study pollutant dispersion and chemistry.  The model is driven by meteorological fields and the parameterization of SBL height as a function of the near-surface temperature inversion strength. The results of this model are in excellent agreement with both surface mixing ratios of $SO_2$ and even vertically resolved measurements over this full field study.  They are clearly highly useful and meaningful for understanding pollution levels and how environmental variables affect that pollution.

The reviewer seems to think that the model predicts temperature or other meteorological variables, while in fact these are inputs to the model and are not predicted online.  The sensitivity study demonstrates that the assumed value of u\* has only a weak effect on the simulation results (see Table S1 in the supplement). The value of u\* may be enhanced by the presence of the rough urban canopy, leading to a value u\* within the canopy (where most $SO_2$ resides) that does not fit with the approximate relationship that the reviewer refers to.  However, it is important to remember that our use of u\* is only a parameter to calculate the Kz profile, which is all the model uses.  Our study directly observes times when $CO_2$ is at regional background levels at the roof of the CTC building and there is more than 100 ppm more $CO_2$ at the ground level.  Clearly the air in contact with the ground is not vertically mixing to the building top (23m AGL) and showing the polluted layer height is ~20m at times, and is not 280m as their

expression would give. As discussed earlier and is discussed in Anderson and Neff (2008), the formation of the temperature structure can be a longer term process. The reviewer indicates a scaling relationship for the *equilibrium* (e.g., formed at very long time) SBL height that would predict a taller SBL height than was directly observed through tracer dispersion in this study. Since the reviewer appears to be concerned with the temperature structure, which is formed on longer timescales or through larger scale weather processes (e.g., warm air advection), it is reasonable that the chemically relevant tracer SBL height on the timescale of pollution export is as we observe here (20-50m), while the temperature-based BL height could be significantly higher and is formed on a longer timescale.

*Reproducibility: I have difficulties following the single column model setup. As a single column model user, I need a roughness length (for momentum, heat and species), geostrophic wind (in height and time) speed, initial profiles, timestep, subsidence rate, advection of momentum, heat, and moisture. These are not reported. Perhaps it is good have a look at the GABLS3 model setup paper (Bosveld et al, 2014) that discusses all these forcing in detail. Moreover I miss a detailed model evaluation against the meteorological fields to show the model is trustworthy.*

Again, this comment demonstrates a misunderstanding of the manuscript. The model in the present manuscript is not like the model the reviewer is describing. The PACT-1D model is described in these published papers (Tuite et al., 2021 and Ahmed et al., 2022) and the model code was provided to the reviewer and will be made public upon acceptance of the manuscript. Because PACT-1D is open source, the reproducibility of the model is not an issue. The model is not "free running" and doesn't simulate the boundary layer structure from the parameters that the reviewer describes. It is a dispersion model that is validated against independently measured tracer ($SO_2$ and $CO_2$) concentration profiles, where it shows excellent skill. Since the model uses the meteorological observations as input, we cannot evaluate model results against those same observations.

*Writing Style and organization. As said before the paper is too long, the Introduction is too long and not too the point. In the results section many paragraphs start with phrases like "Figure X shows a time series of....". In terms if style it is not attractive. But also these sentences belong to the Figure caption, not the main text. In terms of organization, the paper first discusses some modelling (the academic cases), then it discusses in detail the observations and then it returns to the modelling. This is somewhat confusing. The conclusion section needs to be more concise and should focus on answering the well-defined research question in the Introduction Section. Sorry I cannot be more positive.*

The writing was intended to be more accessible to chemists and regulators, but we will try to make it shorter and more precise. We chose the order of showing the idealized model, which is relevant to any stable boundary layer situation, not just this specific field study, before starting into the field study results. We will consider the best ordering, but will also make sure that the transitions are clearer such that it is not confusing.

*Minor remarks*

*Ln 1: title is very long and not appealing*

We believe that assertion driven titles, as the one we chose for this article, are helpful to potential readers. They more clearly convey the purpose and main statement of the manuscript than a more generic and shorter title. In addition to pollution researchers and atmospheric scientists, an additional audience of this work is regulators and local residents concerned about air quality problems. Local air quality regulators commended the choice of the title. During revision, we will consider if an alternative title would be more appropriate.

*Ln 64: 40 nanomole/mole. Is that concentration a problem?*

As the text says, this level is below US EPA short term (1 h) standards of 75 nanomole mole$^{-1}$. Although the US doesn't have a lower long-term standard, the Canadian long-term $SO_2$ standard is 5 nanomoles mole$^{-1}$, so it is clear that Fairbanks $SO_2$ levels are of concern.

*Ln 135: Obukhov length? Should it not be the local Obukhov length, as was learnt from Nieuwstadt 1984?*

We can make this substitution.

*Ln 142: At this stage it is unclear how the SBL height is formally defined (based on TKE, inversion strength, LLJ height …)*

On line 131, we define "the SBL height, $h$, which is the height above which the atmosphere is no longer influenced by contact with the surface." This definition is equivalent to the AMS definition of the boundary layer: "The bottom layer of the troposphere that is in contact with the surface of the earth." The parameterization (Table 1) determines the SBL height based upon the current surface temperature gradient. The SBL height is not diagnosed from some other meteorological aspect of the model. The eddy diffusivity (Kz) profile is then generated from this SBL height. Since the Kz profile controls vertical mixing and goes to very low values above the SBL height, the air above the SBL height is decoupled from the surface. Because the temperature inversion strength varies in time, the parameterized SBL height varies in time. We also find observationally that when there is an extremely strong temperature inversion, the SBL height goes below the top of the CTC building and we observe regional background $CO_2$ mixing ratios on the roof (23m AGL) even though ground-level air (3m AGL) is highly polluted. These direct observations confirm that we are measuring the polluted SBL height.

*Ln 177: co-added: vague. What did you precisely do?*

There are 9 WRF/CMAQ emissions grid cells in the model's footprint. The ADEC/EPA emissions inventory reports an emission rate (tons $SO_2$ per hour). We added the $SO_2$ emissions from these 9 horizontal grid cells that reside in the 1-D model's horizontal footprint and then divided by the area of the model footprint (16 km$^2$) to get the emissions in column density units

(moles m$^{-2}$ s$^{-1}$) to put into the model. This was described more fully in Section 3.4, starting on line 292. For clarity, we will add a reference to Section 3.4 near line 177.

*Ln 225: The behaviour of 1D models depends on the first model height. Please report where it is?*

As described in the published model references (Tuite et al., 2021; Ahmed et al., 2022), the spacing near the surface is logarithmic, having very shallow layers (1mm) at the ground to allow for modeling of deposition's role on atmospheric chemistry. However, in this application, SO$_2$ deposition is considered to be slow and is not included. To improve clarity, we will add to line 225 of the manuscript a description of the logarithmic scale and give the PACT-1D model references.

*Ln 242: Obukhov length. It is Monin-Obukhov theory, but the Obukhov length scale.*

We can make this substitution.

*Ln 288: comparison with sonde data. Please quantify the scores. Models do have difficulties with forecasting SBLs, so "good agreement" is somewhat handwavy.*

The WRF modeling effort was performed by US EPA researchers and provided to us as a part of the ALPACA project. There is no feedback between the WRF model and the model in this study. We use observational winds near the ground and only use the WRF-simulations for wind speeds at layers higher than our observations. The WRF simulations using observational nudging achieved RMS temperature errors of 2.6 K at various downtown sites and including multiple measurements on towers including the measurements made up to 23m on the CTC building. As the reviewer notes, it is difficult for models (here WRF) to predict SBL temperature gradients, so we only used the observations (up to 23m) to quantify the inversion strength.

*Ln 357: conceptual model should be part of the Introduction*

The introduction should include only knowledge before this work was done, so the results of this simulation for idealized cases do not fit the definition of introductory materials.

*Ln 373: Fig 4 is referred to before Figure 3.*

The reviewer is incorrect. On line 369, Figure 3 is first referred to, before line 373.

*Ln 453: 3-h averaging is very coarsening the data, and will help to raise the correlation. Is this fair?*

This averaging time was chosen to compare the in-situ observations to the path-averaged and model results. Due to the intermittent nature of turbulence and proximity to variable influence from sources that are on length scales much smaller than the 4km x 4km footprint of the model,

we decided that it would be best to average to a timescale that is commensurate with transport on the ~4km length scale and was chosen to be 3 hours for reporting statistical agreement of the model.  Note that the LP-DOAS averages on that length scale by its path averaged nature.  The finding of the model that the typical export time from the modeled column is a couple hours also supports this choice of averaging time for statistical comparison.

To address the reviewer's concerns, we can also report 1h statistics.  On the 1h timescale, we find that the 1h correlation coefficients is R=0.83 for all LP-DOAS paths (it was R=0.88 at 3h averaging), and R=0.77 for the 3m in-situ $SO_2$ comparison (was R=0.81 at 3h averaging).  These results are slightly better at 3h than 1h, but are quite close.  Therefore, we feel that the choice made based upon the horizontal length scale and export timescale is valid.

*Figure 1: I miss a scale bar and a north-arrow*

We will add a scale bar and a north arrow.

References:

Ahmed, S., Thomas, J., Tuite, K., Stutz, J., Flocke, F., Orlando, J., Hornbrook, R., Apel, E., Emmons, L., Helmig, D., Boylan, P., Huey, G., Hall, S., Ullmann, K., Cantrell, C., and Fried, A.: The Role of Snow in Controlling Halogen Chemistry and Boundary Layer Oxidation During Arctic Spring: A 1D Modeling Case Study, J. Geophys. Res. Atmospheres, 127, e2021JD036140, https://doi.org/10.1029/2021JD036140, 2022.

Anderson, P. S. and Neff, W. D.: Boundary layer physics over snow and ice, Atmos. Chem. Phys., 8, 3563–3582, https://doi.org/10.5194/acp-8-3563-2008, 2008.

Tuite, K., Thomas, J., Veres, P., Roberts, J., Stevens, P., Griffith, S., Dusanter, S., Flynn, J., Ahmed, S., Emmons, L., Kim, S., Washenfelder, R., Young, C., Tsai, C., Pikelnaya, O., and Stutz, J.: Quantifying Nitrous Acid Formation Mechanisms Using Measured Vertical Profiles During the CalNex 2010 Campaign and 1D Column Modeling, J. Geophys. Res. Atmospheres, 126, e2021JD034689, https://doi.org/10.1029/2021JD034689, 2021.

*-- Reviewer 2 --*

*The manuscript describes measurements and simple modeling of pollutant accumulation in Fairbanks.  It has some strengths, and significant weaknesses.  On balance I think it is not original or novel enough to be publishable in a first-rate scientific journal.*

*Strengths:  An interesting and unusual dataset, with measurements well suited to the problem.  For example, the finding that the concentrations measured by the tower were fairly representative of the long-path is a nice contribution.  The measurements should be published, which does not seem to have happened yet.*

*Weaknesses: The main weakness is that the conclusions are already well known. Trapping of pollutants in a shallow boundary layer in a basin is obvious. The relationship between inversion strength and SBL depth is also not new. The 1D model is not particularly novel either. Such models have been used for decades, why develop another one? If the aim is to forecast polluted conditions, that can be done well enough for practical purposes using routine weather forecasts, informed by local knowledge.*

Reply to reviewer 2:

We find the reviewer's feedback on this manuscript not substantive and note that it has no citations and only qualitative language. We agree with their finding of strengths, but the description of weaknesses are only concepts and the reviewer has not backed them up by literature references. In our reply below, we show, through comparison to published work, that this manuscript makes significant advances from past work, creating a quantitatively accurate parameterization with better performance than prior modeling. While it is true that the concept that strong temperature inversions trap pollution in basins existed, the quantitative degree of that trapping and the extremely shallow mixing height of pollution in Fairbanks wintertime were unknown before this study. No prior study has used a vertically resolved pollution data set to validate a model under such extreme inversion conditions, which we do in this study. We find extremely shallow SBL heights are required to explain the high concentrations of $SO_2$ found near the ground in Fairbanks. As stated by the reviewer, we demonstrate that the lowest path-averaged data agrees well with the in-situ 3m observation, demonstrating a mixed layer near the ground (the urban canopy). The points below describe these findings.

a) The reviewer claims that existing models can model this situation, but we don't know of a paper that presents a model with good skill (small error) in modeling Fairbanks pollution. Mölders et al. (2012) reported 13% normalized mean bias (NMB) and 71% normalized mean error (NME) (for 24-hour PM2.5). We chose to report R in our paper because the fits were so good, but if we were to compare using the same metric as Mölders et al. (2012), we find that our all-paths LP-DOAS $SO_2$ 24-hour statistics are: 25% NMB and 33% NME. Note that our bias is larger, but we have not applied any adjustments to the model and the emissions are only estimated and have significant uncertainties. The NME of our model is half of that of the published model, a huge improvement. By its statistical definition, the NME cannot be smaller than the NMB, so their close values demonstrate that our model is precise, even if slightly positively biased. If we calculate the normalized root-mean-square error (NRMSE) of the model, which is the model RMS error divided by the range of the data (Observed max - min), we find that our model has a LP-DOAS 24-hour NRMSE of 10%, showing excellent prediction precision. Note that the NRMSE is not affected by a bias by the way it is calculated, which is why it is better than the NME.

The past modeling reported 24-hour average statistics, but our work is at higher time resolution allowing us to explore the NRMSE at faster time resolutions. For all LP-DOAS path data and at 24-hour averaging, the RMS error of the model is 2.1 nanomole / mole with a range of 21 nanomole / mole (10% NRMSE). When considering the 1h averaging, the RMSE is 57% larger,

3.3 nanomole / mole, but due to less averaging, the range is now 42.3 nanomole / mole.  This means that while the NRMSE for all LP-DOAS paths is 10% at 24-hour resolution, it *improves* to only 8% error at 1 hour time resolution.  This improvement is because the 24-hour averaging reduces the range of the denominator, which makes the 24-hour statistics worse. Overall, it is clear that our model performs much better, even on a faster time resolution than published models for Fairbanks pollution.

b) While the concept is known that the SBL height is lower when there is a stronger inversion, a quantitative relationship that makes a model work so well was not known before this study.  Past work in Fairbanks (Mayfield and Fochesatto, 2013) measured the "inversion height" from radiosonde temperature data and found an average of 377m for the "surface based inversion". Clearly the observations of $SO_2$ vertical profile and successful modeling of that pollution demonstrate that this "temperature-based surface based inversion" height is not the polluted layer height, and the polluted layer height is much lower than 377m.  This point of differences between temperature-based and tracer-based BL heights was discussed in the response to reviewer 1.

c) Having such a shallow tracer-based BL height is very novel.  To have a rooftop at 23m AGL be above the SBL height (e.g.,not influenced by surface-emitted pollution) is extremely novel. Our observations show that the $CO_2$ at the building top is at regional background levels while the ground-level $CO_2$ sensor was highly polluted.  This directly shows that the air at the top of the building is not "coupled with the surface" e.g., is above the stable boundary layer.  The LP-DOAS $SO_2$ profiles also show that it is trapped on extremely shallow vertical scales.

d) All the other highly stable boundary layer pollution tracer modeling studies we know of compare surface concentrations or mixing ratios, not vertically resolved concentration profiles. Our model performs excellently in both modeling the surface mixing ratio and the profile of the $SO_2$ pollution.  We also show that the 3m in-situ and 15m LP-DOAS data agree well, giving more credence to the LP-DOAS data quality and demonstrating the presence of an "urban canopy." There is also a clear value of developing models that are capable of simulating pollutant dispersion in the vertical (validated with observations at the surface and vertically resolved concentration profiles) to atmospheric chemistry as we discuss below.

e) The reviewer asks "If the aim is to forecast polluted conditions, that can be done well enough for practical purposes using routine weather forecasts, informed by local knowledge."  Our purpose is not forecasting polluted conditions, but a description of the vertical and horizontal dispersion of pollution so that we can determine residence times and in the future how chemical reactions are affected by this poor mixing.  For example, large emissions of NO at ground level quantitatively remove ozone ($O_3$) at ground level, converting NO to $NO_2$ (collectively known as NOx), preventing ozone oxidation reactions at ground level.  However, aloft there is ozone and less NOx, allowing ozone to further oxidize NOx (e.g. forming $NO_3$ and $N_2O_5$) and putting the system into a different chemical regime (Cesler-Maloney et al., 2022).  To address coupled chemical reactions and transport, we need this type of model.  We also note that we are not developing a new model, this model has been used in multiple prior publications (Tuite et al.,

2021; Ahmed et al., 2022) and similar 1-D chemical / transport models have been used in the past (e.g., the MISTRA model). The novelty of this work is adding horizontal export, which allows the understanding of dispersion in this extremely stable multi-day (*e.g.,* not diurnal) inversion situation. We also demonstrate that the extremely shallow SBL heights can be simply parameterized by near-surface vertical temperature gradients.

References:

Cesler-Maloney, M., Simpson, W. R., Miles, T., Mao, J., Law, K. S., & Roberts, T. J., Differences in ozone and particulate matter between ground level and 20 m aloft are frequent during wintertime surface-based temperature inversions in Fairbanks, Alaska. Journal of Geophysical Research: Atmospheres, 127, e2021JD036215. https://doi.org/10.1029/2021JD036215 , 2022.

Mayfield, J. A. and Fochesatto, G. J.: The Layered Structure of the Winter Atmospheric Boundary Layer in the Interior of Alaska, J. Appl. Meteorol. Climatol., 52, 953–973, https://doi.org/10.1175/JAMC-D-12-01.1, 2013.

Mölders, N., Tran, H. N. Q., Cahill, C. F., Leelasakultum, K., and Tran, T. T.: Assessment of WRF/Chem PM 2.5 forecasts using mobile and fixed location data from the Fairbanks, Alaska winter 2008/09 field campaign, Atmospheric Pollut. Res., 3, 108–191, https://doi.org/10.5094/APR.2012.018, 2012.